# Differentiable Sparsity via $D$-Gating:
# Simple and Versatile Structured Penalization

**Chris Kolb[1,2]\*, Laetitia Frost[1], Bernd Bischl[1,2], and David Rügamer[1,2]**
[1]Department of Statistics, LMU Munich
[2]Munich Center for Machine Learning (MCML)

## Abstract

Structured sparsity regularization offers a principled way to compact neural networks, but its non-differentiability breaks compatibility with conventional stochastic gradient descent and requires either specialized optimizers or additional post-hoc pruning without formal guarantees. In this work, we propose $D$-Gating, a fully differentiable structured overparameterization that splits each group of weights into a primary weight vector and multiple scalar gating factors. We prove that any local minimum under $D$-Gating is also a local minimum using non-smooth structured $L_{2,2/D}$ penalization, and further show that the $D$-Gating objective converges at least exponentially fast to the $L_{2,2/D}$–regularized loss in the gradient flow limit. Together, our results show that $D$-Gating is theoretically equivalent to solving the original group sparsity problem, yet induces distinct learning dynamics that evolve from a non-sparse regime into sparse optimization. We validate our theory across vision, language, and tabular tasks, where $D$-Gating consistently delivers strong performance–sparsity tradeoffs and outperforms both direct optimization of structured penalties and conventional pruning baselines.

## 1  Introduction

Sparsity in deep learning models has received considerable attention in recent years. On the one hand, *unstructured* sparsity methods remove individual weights to reduce parameter counts and achieve high compression ratios, but they produce irregular connectivity patterns that are hard to accelerate on standard hardware. On the other hand, *structured* sparsity targets entire groups of parameters, such as neurons [49, 62], convolutional filters [38, 40], or attention heads [12, 42, 53, 60, 71], yielding coarser sparsity patterns that translate directly into reductions in floating-point operations and memory requirements on existing hardware, which results in more efficient deployment of large models [6, 14].

Beyond computational advantages, introducing sparsity can also improve generalization performance [10] and increase model interpretability [20]. Nevertheless, most popular sparsification techniques in deep learning are not based on the non-smooth $L_1$ and $L_{2,1}$ penalties widely used in classical statistics and machine learning [55, 56], but rather constitute iterative pruning and retraining pipelines [3, 20, 34], whose main sparsification mechanism is defined by heuristic pruning criteria like parameter magnitudes. In these methods, the pruning step is decoupled from training, making it difficult to characterize precisely what overall objective is optimized and to provide principled guarantees. Further, the decision space is vast—pruning at initialization [10, 11, 18, 36, 54, 61], after training [38, 40, 41, 69], or sparsification during training [33, 46, 48], each with their own subtleties and tradeoffs [6, 14, 20]—making it cumbersome for practitioners to select a method that balances efficiency, accuracy, and theoretical soundness.

---

\*Author correspondence to `chris.kolb@stat.uni-muenchen.de`

39th Conference on Neural Information Processing Systems (NeurIPS 2025).

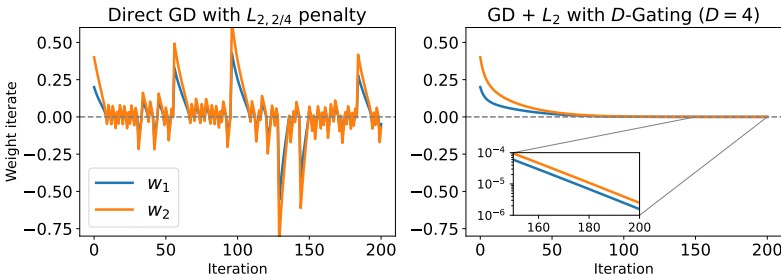

Figure 1: Parameter trajectories for a two-feature squared loss toy objective with non-convex $L_{2,2/D}$ regularization $\mathcal{L}(\mathbf{w}) = (y - x_1 \mathrm{w}_1 - x_2 \mathrm{w}_2)^2 + \lambda \|\mathbf{w}\|_2^{2/D}$ whose global minimizer is $(\mathrm{w}_1^*, \mathrm{w}_2^*) = (0,0)$. **Left**: Failure of direct gradient descent (GD) optimization to converge to $\mathbf{0}$ because of the non-differentiability at the origin. **Right**: $D$-Gated objective where $\mathbf{w} = \boldsymbol{\omega} \cdot \prod_{d=1}^{D-1} \gamma_d$, converging smoothly to $\mathbf{0}$.

**Sparsity penalties** Structured sparsity penalties such as the $L_{2,1}$ norm are, in theory, capable of eliminating unimportant parameter groups, but in deep learning, they have mostly served as heuristics to steer post-hoc pruning rather than achieve exact sparsity [18, 40, 62]. Directly enforcing non-differentiable structured sparsity regularization requires solvers that can cope with its non-smooth nature; if this non-differentiability is ignored, optimization may oscillate or converge to dense, suboptimal solutions, as shown in Fig. 1. Replacing standard stochastic gradient descent (SGD) with specialized procedures such as proximal-type algorithms (e.g. [8, 23, 43]) introduces substantial complexity, demands non-standard hyperparameter configurations, and often the routines are adapted to specific use cases or model classes, thereby foregoing modularity. This renders such approaches cumbersome to implement and inhibits their adoption for large-scale deep learning.

With these obstacles in mind, we ask and positively answer the following main question:

> **Research question**: Can we design a modular structured sparsity regularization method integrable into any architecture, amenable to SGD, with theoretical guarantees and little practical overhead?

## 1.1 Related literature

**Structured sparsity** A range of methods has been proposed to induce block-wise zeros in neural networks, yet they often rely on either post-hoc pruning or non-standard optimizers. Early work applies the convex $L_{2,1}$ penalty directly to weight groups but optimizes it with vanilla (sub-)gradient descent [49, 62], which fails to find sparse solutions and must be followed by an explicit pruning step [8]. To remedy the non-differentiability at zero, [8] proposes the use of a proximal algorithm, which we aim to avoid in favor of compatibility with SGD. [5] further generalizes the $L_{2,1}$ regularizer to non-convex $L_{2,q}$, $q < 1$, penalties using a custom optimizer. Rather than penalizing weight structures directly, [4, 23, 40] introduce shared scaling factors for each group and impose sparsity on those factors instead of the whole parameter group, but still require careful tuning. Although these competitors can yield exact sparsity under certain settings, they either fall back on pruning or abandon standard SGD, motivating our search for a fully differentiable, SGD-compatible alternative.

**Differentiable sparsity** A possible solution to incorporate sparsity-inducing penalties while retaining differentiability are approaches that split the parameters into multiple components and impose smooth $L_2$ regularization on the factors, which can be shown to induce the desired non-smooth sparsity penalty on the reconstructed parameters. This idea dates back to [15, 21] and has recently been adopted to incorporate differentiable $L_1$-type sparsity regularization in neural networks [25, 30, 28, 29, 57, 72, 73]. In the case of matrix (instead of parameter) factorization using two factors, a low-rank bias, given by the trace norm, is induced on the product [27, 52]. The implicit bias literature also investigates such parameter decompositions *without* $L_2$ regularization, which can also induce sparsity under certain conditions–such as impractically small initialization scales [16, 58, 63, 70]. An extension to implicit group sparsity for linear models is presented by [39]. However, existing proposals either focus on unstructured sparsity, are constrained to only two factors, or do not constitute modular approaches applicable to arbitrary architectures. This leaves a gap in the current literature on whether extensions to arbitrary structures are possible, how such an approach can be implemented in practice, and to what extent there are theoretical guarantees to back this method.

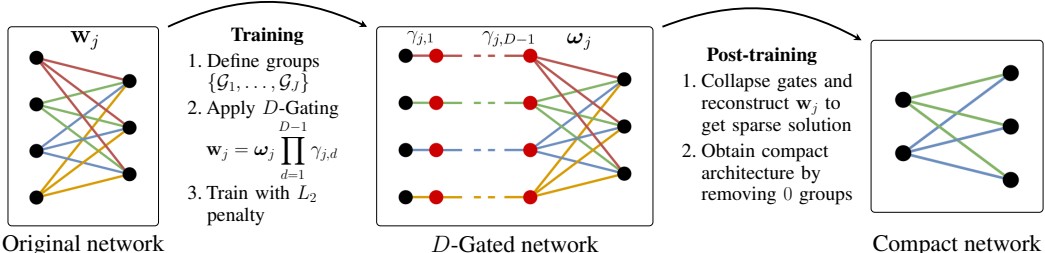

Figure 2: Overview of differentiable $D$-Gating method for structured sparsity (cf. Algorithm 1). For simplicity, we show $D$-Gating visually for a single fully-connected layer with input-wise grouping (colors), but our approach extends to arbitrary network components such as convolutional filters or attention heads. We proceed by applying $D$-Gating (red nodes and their connections) to the neural network weight and running SGD on the gating parameters with weight decay. After training, the weights are collapsed again and the zero structures removed, with the resulting sparse minimizers also being minimizers of the non-smooth $L_{2,2/D}$-regularized objective.

## 1.2 Our contributions

Inspired by prior work on differentiable sparse regularization, we propose a new approach called $D$-Gating, which constitutes a structured sparsity-inducing penalty approach. It can be modularly incorporated in "arbitrary" architectures and neither incurs a notable overhead in additional parameters nor requires additional pruning steps, and is compatible with off-the-shelf SGD optimization. We further establish novel theoretical results that show the equivalence of our proposed differentiable regularization method and non-differentiable sparsity-inducing penalties (cf. Fig. 2), akin to what has been shown for approaches with unstructured sparsity penalties. Apart from theoretically and practically studying the loss landscape and training dynamics of our approach, we also validate our theory on an array of experiments to showcase its versatility in diverse deep learning applications.

## 2 Problem statement

In this paper, we propose a general structured sparsity approach for neural networks $f(\cdot, \mathbf{w})$ that allows penalizing excessive network components without placing restrictions on the type of architecture $f$ or the position of the unit within the weight vector $\mathbf{w} \in \mathbb{R}^p$ that is targeted with the regularization. Structures such as filters naturally arise in neural networks, yielding a partition $\mathcal{G} = \{\mathcal{G}_1, \ldots, \mathcal{G}_J\}$ of a subset of the indices $[p] := \{1, \ldots, p\}$ of $\mathbf{w}$ into $J$ groups $\mathbf{w}_{\mathcal{G}_j}$ with elements $\mathrm{w}_{j,g}, g \in \mathcal{G}_j$. For filter sparsity in convolutional neural networks, $\mathcal{G}$ would be all indices for weights in the convolutional layers, and each $\mathcal{G}_j$ the indices of weights of one of the $J$ filters.

Given this structure, we seek to optimize a general optimization problem

$$\underset{\mathbf{w} \in \mathbb{R}^p}{\text{minimize}} \qquad \mathcal{L}_{\mathbf{w}}(\mathbf{w}) := \mathcal{L}_0(\mathbf{w}) + \lambda \|\mathbf{w}\|_{2,2/D}^{2/D} \qquad (1)$$

for $D \geq 2$, where the unregularized objective $\mathcal{L}_0 = \sum_{i=1}^n \ell(y_i, f(\mathbf{x}_i, \mathbf{w}))$ is the sum of $n$ observed loss contributions with loss $\ell : \mathcal{Y} \times \mathcal{Y} \to \mathbb{R}_0^+$ evaluated on independent data points $\{(\mathbf{x}_i, y_i)\}_{i=1}^n \in (\mathcal{X} \times \mathcal{Y})^n$. The regularization term in Eq. (1) constitutes a generalization of what is often referred to as the group lasso [67], or $L_{2,1}$ penalty, which is recovered for $D = 2$. For $D > 2$, we obtain the more general and non-convex group penalty: $\|\mathbf{w}\|_{2,2/D}^{2/D} := \sum_{j=1}^J (\sum_{g \in \mathcal{G}_j} |\mathrm{w}_{j,g}|^2)^{1/D}$ [22]. As this penalty is neither differentiable for $D = 2$ nor $D > 2$, SGD optimization of (1) yields unfavorable optimization dynamics and does not achieve exact sparsity (cf. Fig. 1). We therefore either require specialized optimization routines or a surrogate objective which induces the solution to (1). We choose the latter to minimize the overhead (cf. App. E.2) and changes to established training procedures.

## 3 Differentiable structured sparsity via $D$-Gating

To solve Eq. (1) while enabling practitioners to use standard SGD optimizers, we derive a fully differentiable method that implicitly tackles Eq. (1) by employing an overparameterized model and a smooth surrogate penalty compatible with SGD optimization.

## 3.1 Model structure and gating variables

As we are interested in a general method for structured sparsity where arbitrary subsets of network weights can be sparsified, we assume a parametric learning model

$$f : \mathcal{X} \times \mathbb{R}^{q+p} \to \mathbb{R}^c, \quad (\mathbf{x}, \widetilde{\mathbf{w}}) \mapsto f(\mathbf{x}; \widetilde{\mathbf{w}}), \tag{2}$$

with inputs $\mathbf{x} \in \mathcal{X}$ and parameters $\widetilde{\mathbf{w}} = (\mathbf{v}, \mathbf{w})$, where $\mathbf{w} \in \mathbb{R}^p$ contains the penalized parameters of interest and $\mathbf{v} \in \mathbb{R}^q$ the remaining parameters. Further, $\mathbf{w}$ is a partitioned (structured) weight object comprising $J$ groups, $\mathbf{w} = (\mathbf{w}_j)_{j=1}^J \in \mathbb{R}^{p_1+\ldots+p_J} = \mathbb{R}^p$.

To convert the non-smooth optimization problem into a smooth optimization problem, we subdivide $\mathbf{w}$ into two parts using the following gating operation:

**Definition 1** ($D$-Gating)**.** Let $\mathbf{w} \in \mathbb{R}^p$ and $\mathcal{G} = \{\mathcal{G}_1, \ldots, \mathcal{G}_J\}$ be a partition of the indices $[p]$ of $\mathbf{w}$ into $J$ groups. Further let $\boldsymbol{\omega} \in \mathbb{R}^p$ be the primary weight of the same size as $\mathbf{w}$, $\boldsymbol{\gamma}_d \in \mathbb{R}^J$ one of $D-1$ vectors containing group-wise gating factors, and let $\boldsymbol{\gamma}^{\odot} := \boldsymbol{\gamma}_1 \odot \ldots \odot \boldsymbol{\gamma}_{D-1}$ denote the element-wise product of the gating factors with entries $\gamma_j^{\odot} = \prod_{d=1}^{D-1} \gamma_{j,d}$ for $j \in [J]$. For brevity, we collect the scaling factors in the matrix $\boldsymbol{\Gamma} = [\boldsymbol{\gamma}_1, \ldots, \boldsymbol{\gamma}_{D-1}] \in \mathbb{R}^{J \times (D-1)}$. The $D$-Gating operation $\blacktriangleright : \mathbb{R}^p \times \mathbb{R}^{J \times (D-1)} \to \mathbb{R}^p, (\boldsymbol{\omega}, \boldsymbol{\Gamma}) \mapsto \boldsymbol{\omega} \blacktriangleright \boldsymbol{\gamma}^{\odot}$, decomposes $\mathbf{w}$ as

$$\mathbf{w} = \boldsymbol{\omega} \blacktriangleright \boldsymbol{\gamma}^{\odot} := \left( \boldsymbol{\omega}_j \prod_{d=1}^{D-1} \gamma_{j,d} \right)_{j=1}^J = \left( \boldsymbol{\omega}_j \cdot \gamma_j^{\odot} \right)_{j=1}^J, \tag{3}$$

and we call $\mathbf{w}$ $D$-Gated if it is parametrized as $\boldsymbol{\omega} \blacktriangleright \boldsymbol{\gamma}^{\odot}$.

Intuitively, the $D$-Gating operation splits each group weight $\mathbf{w}_j$ into $D$ factors: the vector $\boldsymbol{\omega}_j$, corresponding to the original group weights, and the $D-1$ additional gating factors $\gamma_{j,d} \in \mathbb{R}$, which are applied multiplicatively to all entries of $\boldsymbol{\omega}_j$.

## 3.2 Differentiable penalty

Given the gated formulation, we can now impose surrogate $L_2$ regularization on $(\boldsymbol{\omega}, \boldsymbol{\Gamma})$, defined as

$$\mathcal{L}(\mathbf{v}, \boldsymbol{\omega}, \boldsymbol{\Gamma}) = \mathcal{L}_0(\mathbf{v}, \boldsymbol{\omega} \blacktriangleright \boldsymbol{\gamma}^{\odot}) + \lambda \, \mathcal{R}(\boldsymbol{\omega}, \boldsymbol{\Gamma}) \tag{4}$$

$$= \sum_{i=1}^n \ell\left(y_i, f\left(\mathbf{x}_i, (\mathbf{v}, \boldsymbol{\omega} \blacktriangleright \boldsymbol{\gamma}^{\odot})\right)\right) + \frac{\lambda}{D} \underbrace{\sum_{j=1}^J \left( \|\boldsymbol{\omega}_j\|_2^2 + \sum_{d=1}^{D-1} \gamma_{j,d}^2 \right)}_{\|\boldsymbol{\omega}\|_2^2 + \|\boldsymbol{\Gamma}\|_F^2} \tag{5}$$

where $\mathcal{L}_0(\mathbf{v}, \mathbf{w})$ denotes the unregularized, differentiable loss function with per-sample loss $\ell$. In the following, we will denote local minimizers of the $D$-Gating objective as

$$(\hat{\mathbf{v}}, \hat{\boldsymbol{\omega}}, \hat{\boldsymbol{\Gamma}}) \in \underset{(\mathbf{v}, \boldsymbol{\omega}, \boldsymbol{\Gamma}) \in \mathbb{R}^{q+p+J(D-1)}}{\arg \operatorname{loc} \min} \mathcal{L}(\mathbf{v}, \boldsymbol{\omega}, \boldsymbol{\Gamma}). \tag{6}$$

While the function values of Eq. (4) are not necessarily equal to those of Eq. (1), our next section provides a theoretical guarantee of the equivalence of both objectives with regards to their minima and shows that the difference between both objectives is vanishing at least exponentially fast.

# 4 Theoretical results

Because the presence of ungated parameters $\mathbf{v}$ is inconsequential for our analyses and all results directly carry over, they will be omitted from now on, and we assume for simplicity of exposition that all model parameters are $D$-Gated.

## 4.1 Stationarity condition and loss simplification

The following result establishes that all stationary points of $\mathcal{L}(\boldsymbol{\omega}, \boldsymbol{\Gamma})$ correspond to balanced $D$-Gating parameters. Otherwise, one could continuously perturb the $D$-Gating parameters toward a more balanced configuration without altering the effective parameter $\mathbf{w} = \boldsymbol{\omega} \blacktriangleright \boldsymbol{\gamma}^{\odot}$ while strictly decreasing the $L_2$ penalty $\mathcal{R}(\boldsymbol{\omega}, \boldsymbol{\Gamma})$.

**Lemma 1** (Balancedness at stationary points). *Let $(\boldsymbol{\omega}, \boldsymbol{\Gamma})$ be $D$-Gating parameters satisfying $\mathbf{w}_j = \boldsymbol{\omega}_j \prod_{d=1}^{D-1} \gamma_{j,d}$ for $j \in [J]$. If $(\boldsymbol{\omega}, \boldsymbol{\Gamma})$ is a stationary point of the $L_2$-regularized objective $\mathcal{L}(\boldsymbol{\omega}, \boldsymbol{\Gamma})$ with $\lambda > 0$, then the gating factors are group-wise balanced in the sense that*

$$\|\boldsymbol{\omega}_j\|_2^2 = \gamma_{j,1}^2 = \cdots = \gamma_{j,D-1}^2 = \|\mathbf{w}_j\|_2^{2/D} \quad \forall j \in [J]. \tag{7}$$

Notably, the loss evaluated at balanced parameters simplifies to reveal its sparsity-inducing nature:

**Corollary 1** (Loss simplification at balanced gating parameters). *Let $(\boldsymbol{\omega}, \boldsymbol{\Gamma})$ be balanced $D$-Gating parameters satisfying $\mathbf{w}_j = \boldsymbol{\omega}_j \prod_{d=1}^{D-1} \gamma_{j,d}$ and Eq. (7) for $j \in [J]$. The $D$-Gated objective $\mathcal{L}(\boldsymbol{\omega}, \boldsymbol{\Gamma})$ in Eq. (5) then simplifies to*

$$\mathcal{L}_0(\boldsymbol{\omega} \blacktriangleright \boldsymbol{\gamma}^{\odot}) + \frac{\lambda}{D}(\|\boldsymbol{\omega}\|_2^2 + \|\boldsymbol{\Gamma}\|_F^2) = \mathcal{L}_0(\mathbf{w}) + \lambda \sum_{j=1}^{J} \|\mathbf{w}_j\|_2^{2/D} = \mathcal{L}_0(\mathbf{w}) + \lambda \underbrace{\|\mathbf{w}\|_{2,2/D}^{2/D}}_{:=\mathcal{R}_{\mathbf{w}}(\mathbf{w})} =: \mathcal{L}_{\mathbf{w}}(\mathbf{w}) \tag{8}$$

### 4.2 Equivalence of optimization problems

The previous result is reassuring as it demonstrates the equivalence of objectives at balanced gating parameters. It does, however, not guarantee that optimizing one objective provides a meaningful solution for the other objective. The following result establishes equivalence at the solution level.

**Theorem 1** (Equivalence of $D$-Gating and $L_{2,2/D}$ regularization). *The two optimization problems*

$$\underset{\boldsymbol{\omega} \in \mathbb{R}^p, \boldsymbol{\Gamma} \in \mathbb{R}^{J \times (D-1)}}{\text{minimize}} \quad \mathcal{L}(\boldsymbol{\omega}, \boldsymbol{\Gamma}) := \mathcal{L}_0(\boldsymbol{\omega} \blacktriangleright \boldsymbol{\gamma}^{\odot}) + \frac{\lambda}{D}(\|\boldsymbol{\omega}\|_2^2 + \|\boldsymbol{\Gamma}\|_F^2) \tag{9}$$

$$\underset{\mathbf{w} \in \mathbb{R}^p}{\text{minimize}} \quad \mathcal{L}_{\mathbf{w}}(\mathbf{w}) := \mathcal{L}_0(\mathbf{w}) + \lambda \|\mathbf{w}\|_{2,2/D}^{2/D} \tag{10}$$

*are equivalent in the sense that their local minima are identical. If $(\hat{\boldsymbol{\omega}}, \hat{\boldsymbol{\Gamma}}) \in \arg \operatorname{loc} \min \mathcal{L}(\boldsymbol{\omega}, \boldsymbol{\Gamma})$, then $\hat{\boldsymbol{\omega}} \blacktriangleright \hat{\boldsymbol{\gamma}}^{\odot} = \hat{\mathbf{w}} \in \arg \operatorname{loc} \min \mathcal{L}_{\mathbf{w}}(\mathbf{w})$, and likewise, if $\hat{\mathbf{w}} \in \arg \operatorname{loc} \min \mathcal{L}_{\mathbf{w}}(\mathbf{w})$, then any balanced gating representation $(\hat{\boldsymbol{\omega}}, \hat{\boldsymbol{\Gamma}})$ such that $\hat{\mathbf{w}} = \hat{\boldsymbol{\omega}} \blacktriangleright \hat{\boldsymbol{\gamma}}^{\odot}$ is a local minimizer of $\mathcal{L}(\boldsymbol{\omega}, \boldsymbol{\Gamma})$.*

Specifically, there is a bijective mapping between the local minimizers of $\mathcal{L}_{\mathbf{w}}(\mathbf{w})$ and the equivalence class of local minimizers of $\mathcal{L}(\boldsymbol{\omega}, \boldsymbol{\Gamma})$, resulting in the same effective parameter $\mathbf{w}$.

### 4.3 Optimization dynamics

Under ubiquitous (S)GD-based optimization of the $D$-Gated objective in Eq. (5), as well as its theoretically simpler continuous-time gradient flow (GF) limit with infinitesimal learning rate $\eta$, we can additionally establish results characterizing the evolution of parameter balancedness, i.e., quantify how fast the $D$-Gated objective converges to the original $L_{2,2/D}$ regularized loss.

#### 4.3.1 Evolution of imbalance and loss convergence for $D$-Gated models under GF dynamics

The group-wise continuous-time gradient flow dynamics for $j \in [J]$ are given by

$$\dot{\boldsymbol{\omega}}_j = -\nabla_{\boldsymbol{\omega}_j} \mathcal{L}, \quad \dot{\gamma}_{j,d} = -\partial_{\gamma_{j,d}} \mathcal{L}. \tag{11}$$

The gradients with respect to the $D$-Gating parameters $\boldsymbol{\omega}_j$ and the $\gamma_{j,d}$ are, using the chain rule,

$$\nabla_{\boldsymbol{\omega}_j} \mathcal{L} = \gamma_j^{\odot} \nabla_{\mathbf{w}_j} \mathcal{L}_0 + \frac{2\lambda}{D} \boldsymbol{\omega}_j, \quad \partial_{\gamma_{j,d}} \mathcal{L} = \left(\frac{\gamma_j^{\odot}}{\gamma_{j,d}}\right) \boldsymbol{\omega}_j^{\top} \nabla_{\mathbf{w}_j} \mathcal{L}_0 + \frac{2\lambda}{D} \gamma_{j,d}, \, d \in [D-1]. \tag{12}$$

We define the pair-wise imbalance $\mathcal{I}$ between any two group-wise factors $d \neq d' \in [D]$ and show it vanishes exponentially in time:

$$\mathcal{I}_{j,d,d'}(t) := \begin{cases} \|\boldsymbol{\omega}_j(t)\|_2^2 - \gamma_{j,d'}(t)^2, & \text{if } d = 1, \\ \gamma_{j,d}(t)^2 - \gamma_{j,d'}(t)^2, & \text{if } d \neq 1. \end{cases} \tag{13}$$

**Lemma 2** (Exponential decay of imbalance under continuous-time GF). *Under the gradient flow dynamics Eqs. (11) and (12), the pair-wise imbalance $\mathcal{I}_{j,d,d'}(t)$ (13) between two gating parameters $d, d'$ of group $j$ satisfies*

$$\frac{\mathrm{d}}{\mathrm{d}t} \mathcal{I}_{j,d,d'}(t) = -\frac{4\lambda}{D} \mathcal{I}_{j,d,d'}(t) \; \forall d \neq d', j \in [J]. \tag{14}$$

*Solving the ODE shows $\mathcal{I}_{j,d,d'}(t)$ decays exponentially for $\lambda \geq 0$: $\mathcal{I}_{j,d,d'}(t) = \mathcal{I}_{j,d,d'}(0)e^{-\frac{4\lambda}{D}t}$.*

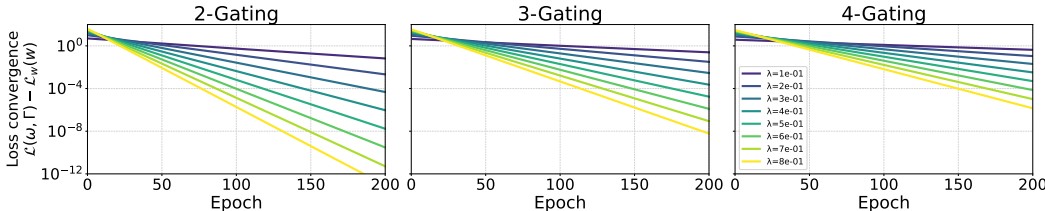

Figure 3: Evolution of imbalance during SGD of a neuron-wise $D$-Gated LeNet-300-100 for $D \in \{2, 3, 4\}$ (**left** to **right**). As predicted by our theory, the losses converge exponentially, with the rate increasing in $\lambda$ and decreasing in $D$.

This result further shows that for $\lambda = 0$, $\mathcal{I}_{j,d,d'}(t)$ is a *conserved quantity* [32, 74], i.e., imbalances decay with a 0 rate. The difference of losses is determined by the difference of regularizers, termed *misalignment* $\mathcal{M}(\boldsymbol{\omega}, \boldsymbol{\Gamma})$, and thus depends on the overall degree of balancedness.

$$\mathcal{L}(\boldsymbol{\omega}, \boldsymbol{\Gamma}) - \mathcal{L}_{\mathbf{w}}(\boldsymbol{\omega} \blacktriangleright \boldsymbol{\gamma}^{\odot}) = \lambda \mathcal{M}(\boldsymbol{\omega}, \boldsymbol{\Gamma}) := \lambda \big( \mathcal{R}(\boldsymbol{\omega}, \boldsymbol{\Gamma}) - \mathcal{R}_{\mathbf{w}}(\boldsymbol{\omega} \blacktriangleright \boldsymbol{\gamma}^{\odot}) \big) \tag{15}$$

$$= \lambda \big( D^{-1}(\|\boldsymbol{\omega}\|_2^2 + \|\boldsymbol{\Gamma}\|_F^2) - \|\boldsymbol{\omega} \blacktriangleright \boldsymbol{\gamma}^{\odot}\|_{2,2/D}^{2/D} \big) \geq 0. \tag{16}$$

Using this, the previous result can be extended to show $|\mathcal{L}(\boldsymbol{\omega}, \boldsymbol{\Gamma}) - \mathcal{L}_{\mathbf{w}}(\boldsymbol{\omega} \blacktriangleright \boldsymbol{\gamma}^{\odot})| \to 0$:

**Lemma 3** (Convergence of $D$-gated loss to $L_{2,2/D}$ regularized loss under GF). *Assume that the model parameters of Eq. (4) with effective weight $\mathbf{w}(t) = \boldsymbol{\omega}(t) \blacktriangleright \boldsymbol{\gamma}^{\odot}(t)$ as in Eq. (3) evolve with time $t$ according to the gradient flow in Eqs. (11) and (12). Then, the $D$-Gated loss $\mathcal{L}(\boldsymbol{\omega}(t), \boldsymbol{\Gamma}(t))$ in Eq. (9) converges to the non-smooth $L_{2,2/D}$ regularized loss $\mathcal{L}_{\mathbf{w}}(\mathbf{w}(t))$ in Eq. (10) at least exponentially fast given an initialization-dependent constant $C \geq 0$:*

$$\mathcal{L}(\boldsymbol{\omega}(t), \boldsymbol{\Gamma}(t)) - \mathcal{L}_{\mathbf{w}}(\mathbf{w}(t)) \leq C e^{-\frac{4\lambda}{D}t}, \tag{17}$$

Intuitively, this is because balancedness in $D$-Gating is precisely the condition required for $\mathcal{L}(\boldsymbol{\omega}, \boldsymbol{\Gamma})$ to simplify to $\mathcal{L}_{\mathbf{w}}(\mathbf{w})$ (cf. Corollary 1). Hence, as the pair-wise imbalances vanish, the balancedness condition becomes increasingly true, and the two losses converge.

### 4.3.2 Evolution of imbalance for $D$-Gated models under (S)GD dynamics

For an analysis of the discrete-time evolution of imbalances, the dynamics becomes more convoluted, but we can establish geometric decay up to first-order in $\eta$, and find symmetry-induced absorbing SGD states [72] for balanced gating configurations.

**Lemma 4** (Imbalance evolution under discrete-time GD). *Consider the $D$-Gated objective Eq. (5). Then, under (S)GD, for any $j \in [J]$, (i) the pair-wise imbalances in Eq. (13) evolve as*

$$\mathcal{I}_{j,d,d'}^{(t+1)} = \big(1 - 4\lambda\eta/D\big) \mathcal{I}_{j,d,d'}^{(t)} + \eta^2 \Delta_{j,d,d'}^{(t)}, \quad \eta > 0, \quad d, d' \in [D], \ d \neq d', \tag{18}$$

*with separate second-order terms $\Delta_{j,d,d'}$ for $d = 1$ and $d, d' > 1$. For sufficiently small $\eta$ or near stationarity, the imbalance $\mathcal{I}_{j,d,d'}^{(t+1)} \approx (1 - 4\lambda\eta/D) \cdot \mathcal{I}_{j,d,d'}^{(t)}$ exhibits discrete exponential decay. (ii) Balancedness is conserved between any two scalar factors $d, d' > 1$, i.e., $\mathcal{I}_{j,d,d'}^{(t)} = 0 \Rightarrow \mathcal{I}_{j,d,d'}^{(t')} = 0 \forall t' > t$, and (iii), for balanced zero representations $(\boldsymbol{\omega}_j^{(t)}, \{\gamma_{j,d}^{(t)}\}_{d=1}^{D-1}) = \mathbf{0}$, it holds $(\boldsymbol{\omega}_j^{(t')}, \{\gamma_{j,d}^{(t')}\}_{d=1}^{D-1}) = \mathbf{0} \forall t' > t$.*

## 5 Numerical experiments

In the following, we empirically investigate the learning dynamics of our approach in Section 5.1 and then showcase various applications in Section 5.2 to demonstrate our method's modularity.

### 5.1 Learning dynamics and misalignment

#### 5.1.1 Exponential decay of imbalance

We first validate our theoretical results on learning dynamics and loss convergence of $D$-Gating from Section 4.3. For this, we apply $D$-Gating with $D \in \{2, 3, 4\}$ to a LeNet-300-100 at the neuron level

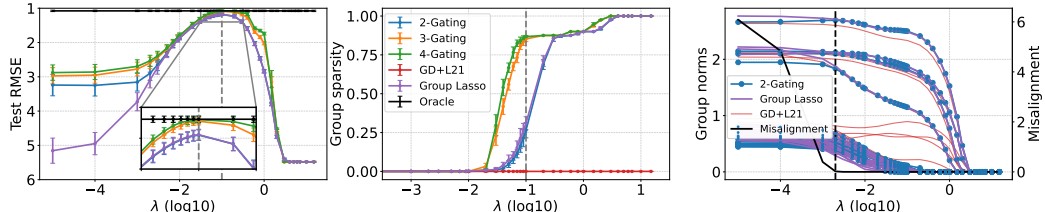

Figure 4: Regularization paths for sparse linear regression task using $D$-Gating. **Left**: Test RMSE vs $\lambda$. The curves for 2-Gating and group lasso coincide beyond a certain $\lambda$, but are outperformed by $D$-Gating with $D > 2$. **Middle**: Group sparsity of 2-Gating coincides with group lasso solution. Dashed grey line indicates optimal $\lambda$ for all models. Deeper gating yields sparser solutions. **Right**: Transition of 2-Gating to group lasso solution beyond a certain $\lambda$ coincides with zero imbalance attained after training. Direct optimization with GD yields notably different regularization paths far from the global minima. The dashed black line indicates $0$ misalignment at the end of traininig. Means and $95\%$ confidence intervals over ten simulations are shown for the left two plots.

and train the model on the MNIST dataset using SGD. We use a grid of $\lambda$ values and measure the loss convergence as defined in Eq. (17). Fig. 3 visualizes the results and confirms our theoretical findings on the exponential decay of the loss difference. Appendix E.4 contains further results, e.g., for Adam.

### 5.1.2   $D$-Gating and misalignment for group lasso

To further validate the equivalence of optimization problems as established in the previous section, we run a sparse linear regression where direct $L_{2,1}$-regularized optimization is more accessible due to the availability of specialized optimization routines. For this, we simulate data as described in Appendix C.1 with 40 feature groups of 5 features each, of which 7 are informative (with truly non-zero effects). We use accelerated proximal gradient descent [50] to directly optimize the original $L_{2,1}$-penalized linear model and apply our approach for $D \in \{2, 3, 4\}$ over the same grid of $\lambda$ values. In addition, we also perform direct GD optimization of the $L_{2,1}$-regularized linear model and compare all methods against an oracle (a linear model using only the signal variables). Results in Fig. 4 confirm the established equivalence between the original and $D$-Gated objective for $D = 2$, but also demonstrate the improvement in the performance-sparsity tradeoff for $D > 2$.

## 5.2   Modularity

Next, we demonstrate the flexibility of our method. To this end, we study various types of structured sparsity problems that arise in neural networks. In these experiments, we focus on demonstrating the broad applicability of our method rather than an exhaustive benchmark comparison. Our method

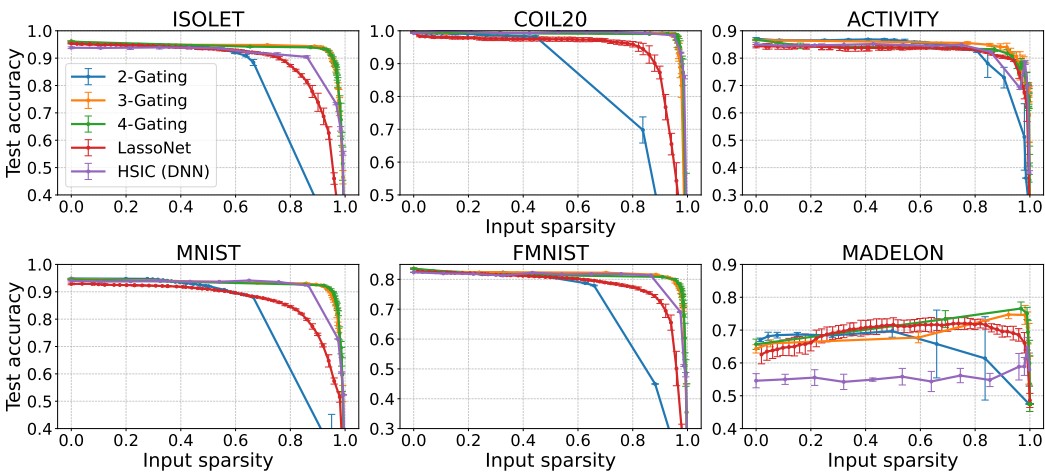

Figure 5: Comparison of feature selection methods. Means and std. over 5 random initializations are reported.

supports any form of structured sparsity in neural networks, enabling diverse applications. Selected use cases are shown below. Further results are presented in Appendix E.

### 5.2.1 Feature selection in non-linear models

We start by investigating input feature selection, i.e., by individually gating the first-layer weights outgoing from each input feature. We follow the setup of [37] by running the proposed method, *LassoNet*, as well as *HSIC* [66] on six diverse datasets as done in [37]. We use the same LeNet-300-100 architecture as a backbone for LassoNet, HSIC and our $D$-Gating approach (cf. Appendix C.2).

Fig. 5 depicts the comparison results, showing that 2-Gating is often inferior to LassoNet and HSIC. 3- and 4-Gating, however, dominate all other methods for almost all possible input sparsity configurations and, hence, are the favorable options among these methods for feature selection.

### 5.2.2 Filter sparsity in convolutional neural networks

In the next experiment, we investigate filter sparsification — one of the most prominent applications of structured sparsity in neural networks. As comparison methods, we select three commonly used methods in the filter sparsity literature [20]: Global magnitude pruning, $L_{2,1}$-penalization with naïve optimization followed by magnitude pruning (MP) [62], and network slimming [40]. A more detailed description can be found in Appendix C.3. We run experiments on CIFAR-10, CIFAR-100, and SVHN, using a VGG-16 [51] and ResNet-18 model [19]. $D$-Gating is implemented by adding gating parameters on the filter level, which, given the size of these models, has a negligible parameter overhead (see Table 5). As filter sparsity can be used to construct a smaller model, potentially deployable on edge devices or similar, we also measure the theoretical speed-up of the sparsified model using floating-point operations (FLOPs). Similar to previous results, Fig. 6 unveils a superior performance of $D > 2$-Gating compared to gating with $D = 2$. However, all gating approaches outperform the filter sparsity baselines despite our approach not requiring post-hoc pruning as the main sparsification mechanism. This is the case both in terms of the accuracy-sparsity tradeoff provided by our method as well as the theoretical speed-up implied (cf. Table 1 and Fig. 11).

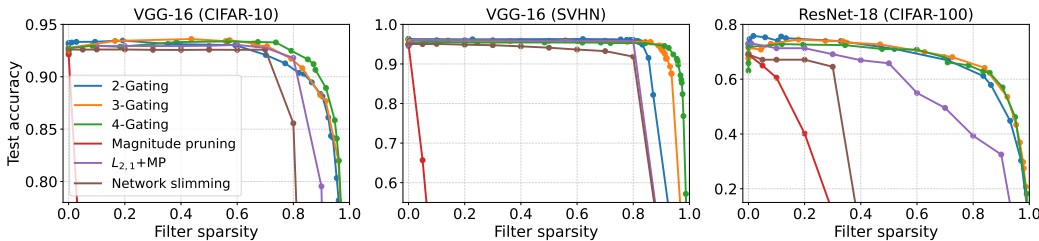

Figure 6: Structured accuracy-sparsity tradeoffs of $D$-Gated neural networks and comparison methods.

Table 1: Largest theoretical speed-up ($\frac{\text{base}}{\text{sparse}}$ FLOPs) within $5\%$ / $10\%$ of the max. test accuracy.

| Method | VGG-16 (CIFAR-10) | VGG-16 (SVHN) | ResNet-18 (CIFAR-100) |
|---|---|---|---|
| D-Gating ($D = 2$) | 16.13 / 52.70 | 175.49 / 175.49 | 5.13 / 14.91 |
| D-Gating ($D = 3$) | 11.17 / 25.51 | 266.66 / 390.03 | 8.27 / 15.14 |
| D-Gating ($D = 4$) | 18.88 / 48.97 | 262.05 / 541.65 | 12.76 / 41.04 |
| $L_{2,1}$ + Mag. pruning | 6.97 / 6.97 | 18.88 / 18.88 | 3.35 / 14.01 |
| Network Slimming | 2.45 / 4.91 | 7.53 / 7.53 | 3.56 / 3.56 |

### 5.2.3 Structured sparsity in language modeling

Our next application considers the effect of $D$-Gating in an attention-based language model [59]. For this, we use NanoGPT and apply $D$-Gating to the attention heads of all attention layers. A natural comparison is again the direct optimization of the $L_{2,1}$-penalty, i.e., without first transforming the objective into a differentiable one through $D$-Gating. To highlight the shortcomings of this naïve approach, we also follow the direct optimization with an explicit pruning step. We train NanoGPT on

TinyShakespeare (details in Appendix C.4) and evaluate the model's validation accuracy as well as validation perplexity for different regularization strengths and hence levels of attention head sparsity.

Fig. 7 confirms our hypothesis that direct optimization does not result in structured sparsity. Notably, the min. and max. norms of the attention heads converge for large $\lambda$ under direct $L_{2,1}$ penalization. In contrast, $D$-Gating shows the desired effect of increased sparsity for higher regularization and provides a smooth tradeoff between accuracy and sparsity. Even when combining direct optimization of the $L_{2,1}$ regularized objective with additional post-hoc pruning, and taking, at each pruning ratio, the best performance over a grid of $\lambda$ values, we see that $D$-Gating achieves much higher head sparsity values before performance degrades significantly.

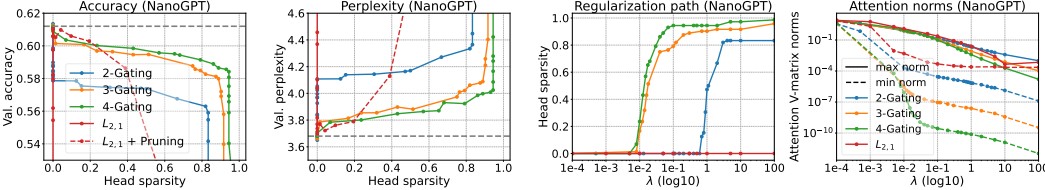

Figure 7: Structured sparsity-performance tradeoffs for NanoGPT trained on TinyShakespeare with $D$-Gated Value matrices in their attention heads and direct $L_{2,1}$ regularization. **Left**: Val. accuracy vs. head sparsity. **Middle**: Val. perplexity vs. $\lambda$. **Right two**: Effect of regularization $\lambda$ on head sparsity and evolution of 2-norms for $D \in \{2, 3, 4\}$. Dashed horizontal lines represent the vanilla performance.

### 5.2.4 Tree sparsity in neural trees

As a final application, we investigate the sparsification of neural trees. More specifically, we propose a novel modification of Neural Oblivious Decision Ensembles (NODE) [47], a neural network-based decision tree ensemble model. While there are multiple options to apply our approach within this architecture, we demonstrate the efficacy of $D$-Gating by inducing sparsity on the tree-level, i.e., using the different trees as groups. We test our approach on the Wine data set [7] using a range of $\lambda$ values for a single tree-layer as suggested by [47]. The tree layer consists of 500 trees, and the weights corresponding to each tree are gated with $D = 2$ to induce differentiable $L_{2,1}$ group sparsity. This is already lower than the default hyperparameter for the tree count of 2048 reported in [47], but raises the question whether 500 trees are in fact necessary to obtain reported performances. Fig. 8 shows the test

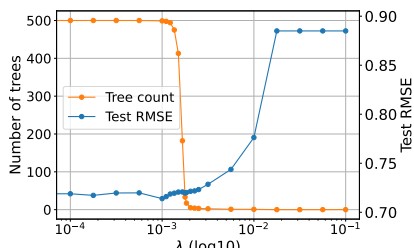

Figure 8: Regularization paths for NODE on the Wine dataset with 2-Gating on tree level. Means and standard deviations over four data splits are shown.

root mean squared error (RMSE) and the number of active trees as functions of $\lambda$. While the full non-sparse model reaches baseline RMSE values, we observe that it is possible to achieve a very similar performance by using less than 5 trees, e.g., for $\lambda = 3 \times 10^{-3}$. This suggests that, at least for the Wine data set, a much simpler and notably less expensive configuration is sufficient.

### 5.2.5 Further experiments and ablation studies

Additional experimental results are provided in Appendix E. Appendices E.5 and E.6 demonstrate the effectiveness of $D$-Gating beyond the model classes studied above. First, $D$-Gating is implemented for multi-modal subnetwork selection in late-fusion architectures, where it consistently succeeds in removing irrelevant data modalities and retaining only informative information.
Next, Appendix E.6 studies a variant of Neural Additive Models (NAMs) [1] with differentiable shape function sparsity, termed $D$-SNAMs. NAMs combine the inherent interpretability of additive models with the expressivity of neural networks by processing each input independently through its own shape function subnetwork before summing the outputs. Here, $D$-Gating is applied to the first-layer weights of each feature-specific subnetwork to enforce shape function sparsity, effectively removing uninformative inputs while maintaining the flexibility to model non-linear effects of informative features. Our differentiable approach outperforms competing methods in terms of

predictive performance, such as sparse NAMs (SNAMs) [65], which are based on non-differentiable (group lasso) penalties, and does not require post-hoc pruning. In particular, we observe for $D > 2$, i.e., non-convex induced regularization, that $D$-Gating produces increasingly sparse solutions while maintaining low generalization error, explainable by the the more aggressive sparsification capabilities of non-convex over convex sparsity penalties [22]. These experiments substantiate $D$-Gating as a promising approach for subnetwork sparsification, which is amenable to differentiable optimization, whether in multi-modal settings or for attaining shape function sparsity in neural additive modeling. Finally, Appendix E.2 includes further information and experiments on the negligible parameter, runtime, and memory overheads incurred by overparameterization using $D$-Gating, while Appendix E.3 contains ablation studies on performance and numerical stability with respect to the gating depth $D$, supporting the recommendation that $D \in \{3, 4\}$ typically yields the best tradeoff.

## 6 Discussion

In this paper, we introduce $D$-Gating, a differentiable structured sparsity method compatible with SGD and applicable to arbitrary differentiable architectures, addressing limitations of non-differentiable penalties in deep learning. We thereby positively answer our initial research question on whether it is possible to design a modular structured sparsity routine, integrable into any architecture, amenable to SGD, with theoretical sparsity guarantees and little practical overhead.

**Limitations and future work**   Due to the flexibility of $D$-Gating, our approach can provide structured sparsity penalties for arbitrary grouping structures. We systematically demonstrate this flexibility through a diverse set of applications in Section 5.2 and Appendix E. While our theoretical results guarantee equivalence to the original sparse but non-smooth optimization problem, future work could further explore the benefits of this formulation or assess its performance when combined with sophisticated pruning and retraining schedules. Secondly, although the gradient flow limit admits clear analysis, it remains an open question how discrete-time SGD with large learning rates or scheduling impacts the learning dynamics. Finally, integrating $D$-Gating into the complex training pipelines of modern large-scale foundation models, where sparse training from scratch is often impractical, presents another promising direction.

## Acknowledgments and Disclosure of Funding

We thank the four anonymous reviewers and the area chair for providing valuable feedback.

DR's and CK's research is funded by the Deutsche Forschungsgemeinschaft (DFG, German Research Foundation) – 548823575.

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

# Appendices

# A Algorithm

In Algorithm 1, we provide the algorithm for sparse training using the proposed $D$-Gating method. Not that it is assumed here that all model parameters $\mathbf{w}$ are gated, which is inconsequential for the procedure. If $D$-Gating is applied only to model substructures, the remaining weights are initialized and updated as usual, and then copied over to the sparse architecture without modification.

---

**Algorithm 1** $D$-Gating Network Training

---

1: **Input:**
2:     Data $\mathcal{D} = \{(\mathbf{x}_i, y_i)\}_{i=1}^n$, model architecture $f$ with weights $\mathbf{w} \in \mathbb{R}^p$ partitioned into $J$ groups $\{\mathcal{G}_j\}_{j=1}^J$
3:     Gating depth $D \geq 2$
4:     Training hyperparameters $\{T, |B|, \{\eta^{(t)}\}_{t=0}^T, \lambda\}$
5:     Threshold $\varepsilon_{\text{tiny}}$
6: **$D$-Gating parametrization:**
7:     Replace weights $\mathbf{w}$ in $f$ by primary weights $\boldsymbol{\omega} = (\boldsymbol{\omega}_j)_{j=1}^J$ and gating factors $\boldsymbol{\Gamma} = (\gamma_{j,d})_{j\in[J],\, d\in[D-1]}$
8:     Effective weights $\mathbf{w} = \boldsymbol{\omega} \blacktriangleright \boldsymbol{\gamma}^{\odot}$, so $\mathbf{w}_j = \boldsymbol{\omega}_j \prod_{d=1}^{D-1} \gamma_{j,d}$

9: **Initialize** $\boldsymbol{\omega}^{(0)} \leftarrow \texttt{Standard-Init}(\mathbf{w}|f)$, $\boldsymbol{\Gamma}^{(0)} \leftarrow \mathbf{1}_J \mathbf{1}_{D-1}^\top \in \mathbb{R}^{J\times(D-1)}$, and $\eta \leftarrow \eta^{(0)}$

10: **for** $t = 0$ to $T-1$ **do**
11:     Sample mini-batch $B^{(t)} \subseteq \mathcal{D}$
12:     Compute $\mathbf{w}^{(t)} = \boldsymbol{\omega}^{(t)} \blacktriangleright (\boldsymbol{\gamma}^{\odot})^{(t)}$ and mini-batch loss

$$\mathcal{L}(\boldsymbol{\omega}^{(t)}, \boldsymbol{\Gamma}^{(t)}) = \sum_{(\mathbf{x},y)\in B^{(t)}} \ell\big(y, f(\mathbf{x}; \mathbf{w}^{(t)})\big) + \frac{\lambda}{D}\Big(\|\boldsymbol{\omega}^{(t)}\|_2^2 + \|\boldsymbol{\Gamma}^{(t)}\|_F^2\Big)$$

13:     Compute gradients $\nabla_{\boldsymbol{\omega}}\mathcal{L}(\boldsymbol{\omega}^{(t)}, \boldsymbol{\Gamma}^{(t)})$, $\nabla_{\boldsymbol{\Gamma}}\mathcal{L}(\boldsymbol{\omega}^{(t)}, \boldsymbol{\Gamma}^{(t)})$
14:     Update

$$\boldsymbol{\omega}^{(t+1)} \leftarrow \boldsymbol{\omega}^{(t)} - \frac{\eta}{|B|}\nabla_{\boldsymbol{\omega}}\mathcal{L}(\boldsymbol{\omega}^{(t)}, \boldsymbol{\Gamma}^{(t)}), \quad \boldsymbol{\Gamma}^{(t+1)} \leftarrow \boldsymbol{\Gamma}^{(t)} - \frac{\eta}{|B|}\nabla_{\boldsymbol{\Gamma}}\mathcal{L}(\boldsymbol{\omega}^{(t)}, \boldsymbol{\Gamma}^{(t)})$$

15:     Set $\eta \leftarrow \eta^{(t+1)}$
16: **end for**

17: **Collapse gates and reduce architecture:**
18:     $\hat{\mathbf{w}} = \boldsymbol{\omega}^{(T)} \blacktriangleright (\boldsymbol{\gamma}^{\odot})^{(T)}$
19:     Zero out removed weight groups: $\hat{\mathbf{w}}_j \leftarrow \mathbf{0}$ if $\|\hat{\mathbf{w}}_j\|_2 < \varepsilon_{\text{tiny}}$
20:     Load sparse weights $\hat{\mathbf{w}}$ into compact architecture $\tilde{f}$
21: **Output:** Sparse network parameters $\hat{\mathbf{w}}$ and reduced architecture $\tilde{f}$

---

# B Missing proofs

## B.1 Proof of Lemma 1

*Proof.* We argue by contradiction. Suppose that for some group $j$ the gating factors are not balanced; that is, there exist indices $d, d' \in [D-1]$ such that, without loss of generality,

$$\gamma_{j,d}^2 < \gamma_{j,d'}^2. \tag{19}$$

An analogous argument applies if one considers an imbalance between $\|\boldsymbol{\omega}_j\|_2^2$ and one of the scalar factors. We now apply a first-order perturbation argument. Consider an infinitesimal perturbation parameter $\varepsilon$ and define the perturbed factors as

$$\tilde{\gamma}_{j,d} = \gamma_{j,d}(1+\varepsilon) \quad \text{and} \quad \tilde{\gamma}_{j,d'} = \gamma_{j,d'}(1-\varepsilon). \tag{20}$$

Noting that

$$(1+\varepsilon)^2 \approx 1 + 2\varepsilon, \quad (1-\varepsilon)^2 \approx 1 - 2\varepsilon, \tag{21}$$

the product of the perturbed factors satisfies

$$\begin{aligned}
\tilde{\gamma}_{j,d}\,\tilde{\gamma}_{j,d'} &= \gamma_{j,d}(1+\varepsilon)\,\gamma_{j,d'}(1-\varepsilon) \\
&= \gamma_{j,d}\gamma_{j,d'}\,(1+\varepsilon)(1-\varepsilon) \\
&= \gamma_{j,d}\gamma_{j,d'}(1-\varepsilon^2) \approx \gamma_{j,d}\gamma_{j,d'},
\end{aligned} \tag{22}$$

so that the effective parameter $\mathbf{w}_j = \boldsymbol{\omega}_j \prod_{d=1}^{D-1} \gamma_{j,d}$ remains unchanged to first order. On the other hand, the original $L_2$ penalty for the two factors is given by

$$\mathcal{R}(\gamma_{j,d}, \gamma_{j,d'}) = \gamma_{j,d}^2 + \gamma_{j,d'}^2. \tag{23}$$

After perturbation, we have

$$(\tilde{\gamma}_{j,d})^2 \approx \gamma_{j,d}^2(1 + 2\varepsilon), \quad (\tilde{\gamma}_{j,d'})^2 \approx \gamma_{j,d'}^2(1 - 2\varepsilon), \tag{24}$$

so that the new penalty becomes

$$\begin{aligned}
\mathcal{R}(\tilde{\gamma}_{j,d}, \tilde{\gamma}_{j,d'}) &\approx \gamma_{j,d}^2(1 + 2\varepsilon) + \gamma_{j,d'}^2(1 - 2\varepsilon) \\
&= \left(\gamma_{j,d}^2 + \gamma_{j,d'}^2\right) + 2\varepsilon\left(\gamma_{j,d}^2 - \gamma_{j,d'}^2\right).
\end{aligned} \tag{25}$$

Since $\gamma_{j,d}^2 < \gamma_{j,d'}^2$ by Eq. (19), it follows that

$$2\varepsilon\left(\gamma_{j,d}^2 - \gamma_{j,d'}^2\right) < 0. \tag{26}$$

This strict decrease in the $L_2$ penalty contradicts the stationarity of $(\boldsymbol{\omega}, \boldsymbol{\Gamma})$ with respect to $\mathcal{L}(\boldsymbol{\omega}, \boldsymbol{\Gamma})$. Therefore, the gating parameters must be balanced at any stationary point, which, together with the invariance of the effective parameter $\mathbf{w}$, implies the result Eq. (7). $\qquad\square$

## B.2   Proof of Corollary 1

*Proof.* We begin by group-wise separating the smooth surrogate penalty in (5):

$$\frac{1}{D}\left(\|\boldsymbol{\omega}\|_2^2 + \|\boldsymbol{\Gamma}\|_F^2\right) = \sum_{j=1}^{J} \frac{1}{D}\left(\|\boldsymbol{\omega}_j\|_2^2 + \sum_{d=1}^{D-1} \gamma_{j,d}^2\right). \tag{27}$$

Next, for each group $j \in [J]$, we apply the inequality of arithmetic and geometric means (AM-GM) to the $D$ non-negative terms $\|\boldsymbol{\omega}_j\|_2^2, \gamma_{j,1}^2, \ldots, \gamma_{j,D-1}^2$:

$$\frac{1}{D}\left(\|\boldsymbol{\omega}_j\|_2^2 + \sum_{d=1}^{D-1} \gamma_{j,d}^2\right) \geq \left(\|\boldsymbol{\omega}_j\|_2^2 \cdot \prod_{d=1}^{D-1} \gamma_{j,d}^2\right)^{1/D}. \tag{28}$$

Since $\mathbf{w}_j = \boldsymbol{\omega}_j \prod_{d=1}^{D-1} \gamma_{j,d}$, the right-hand side becomes

$$\left(\|\boldsymbol{\omega}_j\|_2^2 \cdot \prod_{d=1}^{D-1} \gamma_{j,d}^2\right)^{1/D} = |\prod_{d=1}^{D-1} \gamma_{j,d} \cdot \|\boldsymbol{\omega}_j\|_2|^{2/D} = \|\boldsymbol{\omega}_j \cdot \prod_{d=1}^{D-1} \gamma_{j,d}\|_2^{2/D} = \|\mathbf{w}_j\|_2^{2/D}. \tag{29}$$

Summing over all $J$ groups yields

$$\frac{1}{D}\left(\|\boldsymbol{\omega}\|_2^2 + \|\boldsymbol{\Gamma}\|_F^2\right) \geq \sum_{j=1}^{J} \|\mathbf{w}_j\|_2^{2/D} = \|\mathbf{w}\|_{2,2/D}^{2/D}, \tag{30}$$

with equality holding if and only if $\|\boldsymbol{\omega}_j\|_2^2 = \gamma_{j,1}^2 = \cdots = \gamma_{j,D-1}^2$ for each $j \in [J]$, i.e. exactly the balancedness condition (7). Under that condition, each group-wise penalty attains its minimal value $\|\mathbf{w}_j\|_2^{2/D}$ subject to the constraint that the effective parameter $\mathbf{w}_j$ remains unchanged. Finally, substituting back into $\mathcal{L}(\boldsymbol{\omega}, \boldsymbol{\Gamma}) = \mathcal{L}_0(\boldsymbol{\omega} \blacktriangleright \boldsymbol{\gamma}^\odot) + \frac{\lambda}{D}(\|\boldsymbol{\omega}\|_2^2 + \|\boldsymbol{\Gamma}\|_F^2)$ gives

$$\mathcal{L}(\boldsymbol{\omega}, \boldsymbol{\Gamma}) = \mathcal{L}_0(\mathbf{w}) + \lambda \sum_{j=1}^{J} \|\mathbf{w}_j\|_2^{2/D} = \mathcal{L}_0(\mathbf{w}) + \lambda \|\mathbf{w}\|_{2,2/D}^{2/D} = \mathcal{L}_{\mathbf{w}}(\mathbf{w}), \tag{31}$$

completing the proof. $\qquad\square$

### B.3 Proof of Theorem 1

*Proof.* The two objectives are

$$\mathcal{L}(\boldsymbol{\omega}, \boldsymbol{\Gamma}) = \mathcal{L}_0(\boldsymbol{\omega} \blacktriangleright \boldsymbol{\gamma}^{\odot}) + \frac{\lambda}{D}\big(\|\boldsymbol{\omega}\|_2^2 + \|\boldsymbol{\Gamma}\|_F^2\big), \tag{32}$$

$$\mathcal{L}_{\mathbf{w}}(\mathbf{w}) = \mathcal{L}_0(\mathbf{w}) + \lambda \sum_{j=1}^{J} \|\mathbf{w}_j\|_2^{2/D}. \tag{33}$$

For the first direction, suppose that $\hat{\mathbf{w}}$ is a local minimizer of $\mathcal{L}_{\mathbf{w}}(\mathbf{w})$. Then there is $\varepsilon > 0$ such that
$$\mathcal{L}_{\mathbf{w}}(\hat{\mathbf{w}}) \le \mathcal{L}_{\mathbf{w}}(\mathbf{w}) \ \forall \ \mathbf{w} \in \mathcal{B}(\hat{\mathbf{w}}, \varepsilon), \tag{34}$$
where $\mathcal{B}(\hat{\mathbf{w}}, \varepsilon)$ denotes an $\varepsilon$-ball around $\hat{\mathbf{w}} \in \mathbb{R}^p$. By the multiplicative and surjective nature of the $D$-Gating overparameterization $\blacktriangleright$, we can always choose a *balanced* representation $(\hat{\boldsymbol{\omega}}, \hat{\boldsymbol{\Gamma}})$ in the preimage of $\hat{\mathbf{w}}$, i.e.,

$$\hat{\boldsymbol{\omega}} \blacktriangleright \hat{\boldsymbol{\gamma}}^{\odot} = \hat{\mathbf{w}}, \quad \|\hat{\boldsymbol{\omega}}_j\|_2^2 = \hat{\gamma}_{j,1}^2 = \cdots = \|\hat{\mathbf{w}}_j\|_2^{2/D} \quad \forall \ j = 1, \dots, J. \tag{35}$$

Using Corollary 1, balancedness implies that the $D$-Gated objective simplifies to
$$\mathcal{L}(\hat{\boldsymbol{\omega}}, \hat{\boldsymbol{\Gamma}}) = \mathcal{L}_{\mathbf{w}}(\hat{\mathbf{w}}). \tag{36}$$
Further, by continuity of $\blacktriangleright$, there is $\delta > 0$ such that
$$\|(\boldsymbol{\omega}, \boldsymbol{\Gamma}) - (\hat{\boldsymbol{\omega}}, \hat{\boldsymbol{\Gamma}})\|_2 < \delta \implies \|\boldsymbol{\omega} \blacktriangleright \boldsymbol{\gamma}^{\odot} - \hat{\mathbf{w}}\|_2 < \varepsilon. \tag{37}$$

Hence all $(\boldsymbol{\omega}, \boldsymbol{\Gamma}) \in \mathcal{B}((\hat{\boldsymbol{\omega}}, \hat{\boldsymbol{\Gamma}}), \delta)$ map to some effective weight $\mathbf{w} = \boldsymbol{\omega} \blacktriangleright \boldsymbol{\gamma}^{\odot} \in \mathcal{B}(\hat{\mathbf{w}}, \varepsilon)$. Moreover, the gated objective $\mathcal{L}(\boldsymbol{\omega}, \boldsymbol{\Gamma})$ can be related to its non-differentiable counterpart $\mathcal{L}_{\mathbf{w}}(\mathbf{w})$ as $\mathcal{L}(\boldsymbol{\omega}, \boldsymbol{\Gamma}) = \mathcal{L}_{\mathbf{w}}(\boldsymbol{\omega} \blacktriangleright \boldsymbol{\gamma}^{\odot}) + \lambda \mathcal{M}(\boldsymbol{\omega}, \boldsymbol{\Gamma})$, where $\mathcal{M}(\boldsymbol{\omega}, \boldsymbol{\Gamma}) := D^{-1}(\|\boldsymbol{\omega}\|_2^2 + \|\boldsymbol{\Gamma}\|_F^2) - \|\boldsymbol{\omega} \blacktriangleright \boldsymbol{\gamma}^{\odot}\|_2^{2/D}$ measures the (non-negative) misalignment of the $D$-Gating variables, achieving 0 if and only if $(\boldsymbol{\omega}, \boldsymbol{\Gamma})$ is a balanced representation of the effective weight $\mathbf{w} = \boldsymbol{\omega} \blacktriangleright \boldsymbol{\gamma}^{\odot}$. Then, using Eq. (34) and Eq. (36), we obtain the following chain of inequalities

$$\forall (\boldsymbol{\omega}, \boldsymbol{\Gamma}) \in \mathcal{B}((\hat{\boldsymbol{\omega}}, \hat{\boldsymbol{\Gamma}}), \delta): \mathcal{L}(\boldsymbol{\omega}, \boldsymbol{\Gamma}) = \mathcal{L}_{\mathbf{w}}(\boldsymbol{\omega} \blacktriangleright \boldsymbol{\gamma}^{\odot}) + \lambda \underbrace{\mathcal{M}(\boldsymbol{\omega}, \boldsymbol{\Gamma})}_{\ge 0} \ge \mathcal{L}_{\mathbf{w}}(\boldsymbol{\omega} \blacktriangleright \boldsymbol{\gamma}^{\odot}) \tag{38}$$
$$\ge \mathcal{L}_{\mathbf{w}}(\hat{\mathbf{w}})$$
$$= \mathcal{L}(\hat{\boldsymbol{\omega}}, \hat{\boldsymbol{\Gamma}}),$$

showing $(\hat{\boldsymbol{\omega}}, \hat{\boldsymbol{\Gamma}})$ is a local minimizer of $\mathcal{L}(\boldsymbol{\omega}, \boldsymbol{\Gamma})$.

To show the reverse direction, assume $(\hat{\boldsymbol{\omega}}, \hat{\boldsymbol{\Gamma}})$ is a local minimizer of $\mathcal{L}(\boldsymbol{\omega}, \boldsymbol{\Gamma})$. We can apply Lemma 1 to establish the balancedness of the gating parameters $(\hat{\boldsymbol{\omega}}, \hat{\boldsymbol{\Gamma}})$ and further define the effective weight as $\hat{\mathbf{w}} = \hat{\boldsymbol{\omega}} \blacktriangleright \hat{\boldsymbol{\gamma}}^{\odot}$. Since $(\hat{\boldsymbol{\omega}}, \hat{\boldsymbol{\Gamma}})$ is balanced, $\mathcal{M}(\hat{\boldsymbol{\omega}}, \hat{\boldsymbol{\Gamma}}) = 0$ and thus $\mathcal{L}(\hat{\boldsymbol{\omega}}, \hat{\boldsymbol{\Gamma}}) = \mathcal{L}_{\mathbf{w}}(\hat{\mathbf{w}})$. By local minimality of $(\hat{\boldsymbol{\omega}}, \hat{\boldsymbol{\Gamma}})$, there exists $\delta > 0$ such that

$$\mathcal{L}(\hat{\boldsymbol{\omega}}, \hat{\boldsymbol{\Gamma}}) \le \mathcal{L}(\boldsymbol{\omega}, \boldsymbol{\Gamma}) \quad \forall (\boldsymbol{\omega}, \boldsymbol{\Gamma}) \in \mathcal{B}((\hat{\boldsymbol{\omega}}, \hat{\boldsymbol{\Gamma}}), \delta). \tag{39}$$

Define for each group $j \in [J]$ the continuous function $r_j(\mathbf{w}_j) = \|\mathbf{w}_j\|_2^{1/D}$ on $\mathbb{R}^{p_j}$. Given any $\mathbf{w}' \in \mathbb{R}^p$ in a neighborhood of $\hat{\mathbf{w}}$, we can construct corresponding balanced $D$-Gating parameters as follows:

$$\gamma'_{j,d} = \begin{cases} \text{sign}(\hat{\gamma}_{j,d})\, r_j(\mathbf{w}'_j), & \hat{\gamma}_{j,d} \ne 0, \\ r_j(\mathbf{w}'_j), & \hat{\gamma}_{j,d} = 0. \end{cases} \quad \boldsymbol{\omega}'_j = \begin{cases} \mathbf{w}'_j / \prod_{d=1}^{D-1} \gamma'_{j,d}, & r_j(\mathbf{w}'_j) \ne 0, \\ \mathbf{0}, & r_j(\mathbf{w}'_j) = 0. \end{cases} \tag{40}$$

Then $\boldsymbol{\omega}'_j \prod_{d=1}^{D-1} \gamma'_{j,d} = \mathbf{w}'_j$, and by construction, $\|\boldsymbol{\omega}'_j\|_2^2 = \gamma'^2_{j,d} = r_j(\mathbf{w}'_j)^2 = \|\mathbf{w}'_j\|_2^{2/D}$ for all $j \in [J]$ and $d \in [D-1]$. Let $(\boldsymbol{\omega}', \boldsymbol{\Gamma}')$ denote the collection of group-wise gating parameters, i.e., $(\boldsymbol{\omega}', \boldsymbol{\Gamma}') = (\{\boldsymbol{\omega}'_j\}_{j=1}^J, \{\gamma'_{j,d}\}_{j=1,d=1}^{J,D-1})$. By Corollary 1, we obtain $\mathcal{L}(\boldsymbol{\omega}', \boldsymbol{\Gamma}') = \mathcal{L}_{\mathbf{w}}(\mathbf{w}')$. Continuity of $r_j$ and of the map $\mathbf{w}'_j \mapsto \boldsymbol{\omega}'_j$ guarantees the existence of $\varepsilon > 0$ such that

$$\|\mathbf{w}' - \hat{\mathbf{w}}\|_2 < \varepsilon \implies \|(\boldsymbol{\omega}', \boldsymbol{\Gamma}') - (\hat{\boldsymbol{\omega}}, \hat{\boldsymbol{\Gamma}})\|_2 < \delta. \tag{41}$$

Therefore, for every $\mathbf{w}' \in \mathcal{B}(\hat{\mathbf{w}}, \varepsilon)$, the constructed $D$-Gating parameters $(\boldsymbol{\omega}', \boldsymbol{\Gamma}')$ satisfy Eq. (39) and thus

$$\forall \mathbf{w}' \in \mathcal{B}(\hat{\mathbf{w}}, \varepsilon): \mathcal{L}_{\mathbf{w}}(\mathbf{w}') = \mathcal{L}(\boldsymbol{\omega}', \boldsymbol{\Gamma}') \ge \mathcal{L}(\hat{\boldsymbol{\omega}}, \hat{\boldsymbol{\Gamma}}) = \mathcal{L}_{\mathbf{w}}(\hat{\mathbf{w}}).$$

It follows that $\hat{\mathbf{w}}$ is a local minimizer of $\mathcal{L}_{\mathbf{w}}(\mathbf{w})$, completing the proof. $\qquad \square$

### B.4 Proof of Lemma 2

*Proof.* We differentiate the pair-wise imbalance defined in Eq. (13) with respect to time and treat the two cases of $d = 1$ and $d \neq 1$ separately. In the following we use $\gamma_j^{\odot} \in \mathbb{R}$ to abbreviate $\prod_{d=1}^{D-1} \gamma_{j,d}$.

**Case 1:** For $d = 1$, we have

$$\mathcal{I}_{j,1,d'}(t) = \|\boldsymbol{\omega}_j(t)\|_2^2 - \gamma_{j,d'}(t)^2.$$

Differentiating with respect to time yields

$$\frac{\mathrm{d}}{\mathrm{d}t}\mathcal{I}_{j,1,d'}(t) = 2\,\boldsymbol{\omega}_j(t)^\top \dot{\boldsymbol{\omega}}_j(t) - 2\,\gamma_{j,d'}(t)\,\dot{\gamma}_{j,d'}(t). \tag{42}$$

Substituting the dynamics from Eq. (12) (with the appropriate negative signs) gives

$$\begin{aligned}
\frac{\mathrm{d}}{\mathrm{d}t}\mathcal{I}_{j,1,d'}(t) = {}& -2\left[\gamma_j^{\odot}\,\boldsymbol{\omega}_j(t)^\top \nabla_{\mathbf{w}_j}\mathcal{L}_0 + \frac{2\lambda}{D}\|\boldsymbol{\omega}_j(t)\|_2^2\right] \\
& + 2\,\gamma_{j,d'}(t)\left[\left(\frac{\gamma_j^{\odot}}{\gamma_{j,d'}(t)}\right)\boldsymbol{\omega}_j(t)^\top \nabla_{\mathbf{w}_j}\mathcal{L}_0 + \frac{2\lambda}{D}\,\gamma_{j,d'}(t)\right].
\end{aligned} \tag{43}$$

Since

$$\gamma_j^{\odot} = \gamma_{j,d'}(t)\left(\frac{\gamma_j^{\odot}}{\gamma_{j,d'}(t)}\right),$$

the terms involving $\boldsymbol{\omega}_j(t)^\top \nabla_{\mathbf{w}_j}\mathcal{L}_0$ cancel, leaving

$$\frac{\mathrm{d}}{\mathrm{d}t}\mathcal{I}_{j,1,d'}(t) = -\frac{4\lambda}{D}\left(\|\boldsymbol{\omega}_j(t)\|_2^2 - \gamma_{j,d'}(t)^2\right) = -\frac{4\lambda}{D}\,\mathcal{I}_{j,1,d'}(t). \tag{44}$$

**Case 2:** For $d \neq 1$, we have

$$\mathcal{I}_{j,d,d'}(t) = \gamma_{j,d}(t)^2 - \gamma_{j,d'}(t)^2.$$

Differentiating with respect to $t$ yields

$$\frac{\mathrm{d}}{\mathrm{d}t}\mathcal{I}_{j,d,d'}(t) = 2\,\gamma_{j,d}(t)\,\dot{\gamma}_{j,d}(t) - 2\,\gamma_{j,d'}(t)\,\dot{\gamma}_{j,d'}(t). \tag{45}$$

Substituting the expressions from Eq. (12) for both $\dot{\gamma}_{j,d}(t)$ and $\dot{\gamma}_{j,d'}(t)$ and using the identity

$$\gamma_j^{\odot} = \gamma_{j,d}(t)\left(\frac{\gamma_j^{\odot}}{\gamma_{j,d}(t)}\right),$$

the cancellation of the terms involving $\boldsymbol{\omega}_j(t)^\top \nabla_{\mathbf{w}_j}\mathcal{L}_0$ follows analogously to the first case, yielding

$$\frac{\mathrm{d}}{\mathrm{d}t}\mathcal{I}_{j,d,d'}(t) = -\frac{4\lambda}{D}\left(\gamma_{j,d}(t)^2 - \gamma_{j,d'}(t)^2\right) = -\frac{4\lambda}{D}\,\mathcal{I}_{j,d,d'}(t). \tag{46}$$

Thus, in both cases, we have

$$\frac{\mathrm{d}}{\mathrm{d}t}\mathcal{I}_{j,d,d'}(t) = -\frac{4\lambda}{D}\,\mathcal{I}_{j,d,d'}(t),$$

so that the solution to the differential equation is

$$\mathcal{I}_{j,d,d'}(t) = \mathcal{I}_{j,d,d'}(0)e^{-\frac{4\lambda}{D}t}.$$

$\square$

## B.5 Proof of Lemma 3

*Proof.* For a fixed group $j$, the squared gating parameters are denoted as

$$a_1(t) = \|\boldsymbol{\omega}_j(t)\|_2^2, \quad a_{d+1}(t) = \gamma_{j,d}(t)^2, \ d \in [D-1], \tag{47}$$

the group-level regularizer is $\mathcal{R}_j(t) := \frac{1}{D} \sum_{d=1}^{D} a_d(t)$, and the effective group weight is $\mathbf{w}_j(t) = \boldsymbol{\omega}_j(t) \prod_{d=1}^{D-1} \gamma_{j,d}(t) = \boldsymbol{\omega}_j(t) \cdot \gamma_j^{\odot}(t)$, so that by construction $\|\mathbf{w}_j(t)\|_2^{2/D} = \left(\prod_{d=1}^{D} a_d(t)\right)^{1/D}$. Note that by Lemma 2, for each group $j \in [J]$, the pair-wise imbalances (as in Eq. (13)) satisfy

$$\mathcal{I}_{j,d,d'}(t) = \mathcal{I}_{j,d,d'}(0)e^{-\frac{4\lambda}{D}t}. \tag{48}$$

The overall misalignment in group $j$, measuring the balancedness of the gating representation, is given by

$$\mathcal{M}_j(t) := \mathcal{R}_j(t) - \|\mathbf{w}_j(t)\|_2^{2/D} \geq 0. \tag{49}$$

Let $m_j(t) = \min_{1 \leq d \leq D} a_d(t)$ and $M_j(t) = \max_{1 \leq d \leq D} a_d(t)$. Then, by definition, the maximal pair-wise imbalance for group $j \in [J]$ is $\mathcal{I}_{\max,j}(t) := M_j(t) - m_j(t)$. Since $\mathcal{R}_j(t)$ is an arithmetic mean, it satisfies $m_j(t) \leq \mathcal{R}_j(t) \leq M_j(t)$ and hence is upper bounded by $M_j(t)$. Because the geometric mean is not smaller than its smallest component, it is lower bounded by $m_j(t)$, i.e., $\|\mathbf{w}_j(t)\|_2^{2/D} = \left(\prod_{d=1}^{D} a_d(t)\right)^{1/D} \geq m_j(t)$, so that we have

$$\mathcal{M}_j(t) = \mathcal{R}_j(t) - \|\mathbf{w}_j(t)\|_2^{2/D} \leq M_j(t) - m_j(t) = \mathcal{I}_{\max,j}(t). \tag{50}$$

By Lemma 2, for any pair $(d, d')$ in group $j$ we have

$$|a_d(t) - a_{d'}(t)| \leq |a_d(0) - a_{d'}(0)|e^{-\frac{4\lambda}{D}t}, \tag{51}$$

so that $\mathcal{I}_{\max,j}(t) \leq \mathcal{I}_{\max,j}(0)e^{-\frac{4\lambda}{D}t}$. Hence, $\mathcal{M}_j(t) \leq \mathcal{I}_{\max,j}(0)e^{-\frac{4\lambda}{D}t}$ for every group $j$. Defining $\mathcal{I}_{\max}(0) = \max_{j \in \{1,\ldots,J\}} \mathcal{I}_{\max,j}(0)$, we obtain $\mathcal{M}_j(t) \leq \mathcal{I}_{\max}(0)e^{-\frac{4\lambda}{D}t}$ for all $j$. Summing over $j$, the overall misalignment is

$$\mathcal{M}(t) = \sum_{j=1}^{J} \mathcal{M}_j(t) \leq J \mathcal{I}_{\max}(0)e^{-\frac{4\lambda}{D}t}. \tag{52}$$

By adding and subtracting $\mathcal{R}_{\mathbf{w}}(\mathbf{w}) := \|\mathbf{w}\|_{2,2/D}^{2/D}$ to the $D$-Gated objective $\mathcal{L}(\boldsymbol{\omega}, \boldsymbol{\Gamma})$, we can express it as $\mathcal{L}_{\mathbf{w}}(\mathbf{w})$ plus the misalignment $\mathcal{M}(\boldsymbol{\omega}, \boldsymbol{\Gamma}) \geq 0$, i.e.,

$$\mathcal{L}(\boldsymbol{\omega}, \boldsymbol{\Gamma}) = \mathcal{L}_0(\boldsymbol{\omega} \blacktriangleright \boldsymbol{\gamma}^{\odot}) + \lambda \mathcal{R}(\boldsymbol{\omega}, \boldsymbol{\Gamma}) + \lambda\big(\mathcal{R}_{\mathbf{w}}(\mathbf{w}) - \mathcal{R}_{\mathbf{w}}(\mathbf{w})\big) = \mathcal{L}_{\mathbf{w}}(\boldsymbol{\omega} \blacktriangleright \boldsymbol{\gamma}^{\odot}) + \lambda \mathcal{M}(\boldsymbol{\omega}, \boldsymbol{\Gamma}). \tag{53}$$

As the difference between both losses is simply $\mathcal{L}(\boldsymbol{\omega}(t), \boldsymbol{\Gamma}(t)) - \mathcal{L}_{\mathbf{w}}(\mathbf{w}(t)) = \lambda \mathcal{M}(t)$, it follows that

$$\mathcal{L}(\boldsymbol{\omega}(t), \boldsymbol{\Gamma}(t)) - \mathcal{L}_{\mathbf{w}}(\mathbf{w}(t)) \leq \underbrace{\lambda J \mathcal{I}_{\max}(0)}_{:=C} e^{-\frac{4\lambda}{D}t} = Ce^{-\frac{4\lambda}{D}t}, \tag{54}$$

where $C$ is an initialization-dependent constant. We conclude that as $t \to \infty$, $\mathcal{L}(\boldsymbol{\omega}(t), \boldsymbol{\Gamma}(t))$ converges at least exponentially fast to $\mathcal{L}_{\mathbf{w}}(\mathbf{w}(t))$, with their difference vanishing with rate increasing in $\lambda$ and decreasing in $D$. $\square$

## B.6 Proof of Lemma 4

*Proof.* We first abbreviate the gradient of $\mathcal{L}_0$ w.r.t. $\mathbf{w}_j$ as $\boldsymbol{g}_j^{(t)} := \nabla_{\mathbf{w}_j} \mathcal{L}_0(\mathbf{w}^{(t)})$. For clarity of exposition, w.l.o.g., consider full-batch gradient descent updates, given by

$$\boldsymbol{\omega}_j^{(t+1)} = \boldsymbol{\omega}_j^{(t)} - \eta\Big((\gamma_j^{\odot})^{(t)}\boldsymbol{g}_j^{(t)} + \frac{2\lambda}{D}\boldsymbol{\omega}_j^{(t)}\Big), \ \gamma_{j,d}^{(t+1)} = \gamma_{j,d}^{(t)} - \eta\Big(\prod_{d' \neq d} \gamma_{j,d'}^{(t)} \cdot \boldsymbol{\omega}_j^{(t)\top}\boldsymbol{g}_j^{(t)} + \frac{2\lambda}{D}\gamma_{j,d}^{(t)}\Big), \tag{55}$$

and pair-wise imbalances $\mathcal{I}_{j,1,d'}^{(t)} = \|\boldsymbol{\omega}_j^{(t)}\|_2^2 - (\gamma_{j,d'}^{(t)})^2$ and $\mathcal{I}_{j,d,d'}^{(t)} = (\gamma_{j,d}^{(t)})^2 - (\gamma_{j,d'}^{(t)})^2$ for $d, d' > 1$. We first inspect the vector–scalar case $\mathcal{I}_{j,1,d'}^{(t+1)}$ and start by expanding the squared norm of $\boldsymbol{\omega}_j^{(t+1)}$:

$$
\begin{aligned}
\|\boldsymbol{\omega}_j^{(t+1)}\|_2^2 &= \left\|\boldsymbol{\omega}_j^{(t)} - \eta\Big((\gamma_j^{\odot})^{(t)}\boldsymbol{g}_j^{(t)} + \frac{2\lambda}{D}\boldsymbol{\omega}_j^{(t)}\Big)\right\|_2^2 \\
&= \|\boldsymbol{\omega}_j^{(t)}\|_2^2 - 2\eta(\gamma_j^{\odot})^{(t)}\boldsymbol{\omega}_j^{(t)\top}\boldsymbol{g}_j^{(t)} - \frac{4\lambda}{D}\eta\|\boldsymbol{\omega}_j^{(t)}\|_2^2 + \eta^2\left\|(\gamma_j^{\odot})^{(t)}\boldsymbol{g}_j^{(t)} + \frac{2\lambda}{D}\boldsymbol{\omega}_j^{(t)}\right\|_2^2. \quad (56)
\end{aligned}
$$

Similarly, we expand the squared updated scalar $\gamma_{j,d'}^{(t+1)}$ and separate the terms by their order in $\eta$,

$$
\begin{aligned}
(\gamma_{j,d'}^{(t+1)})^2 &= (\gamma_{j,d'}^{(t)})^2 - 2\eta\gamma_{j,d'}^{(t)} \cdot \prod_{d''\neq d'} \gamma_{j,d''}^{(t)} \cdot \boldsymbol{\omega}_j^{(t)\top}\boldsymbol{g}_j^{(t)} - \frac{4\lambda}{D}\eta(\gamma_{j,d'}^{(t)})^2 \\
&\quad + \eta^2\Big(\prod_{d''\neq d'} \gamma_{j,d''}^{(t)} \cdot \boldsymbol{\omega}_j^{(t)\top}\boldsymbol{g}_j^{(t)} + \frac{2\lambda}{D}\gamma_{j,d'}^{(t)}\Big)^2. \quad (57)
\end{aligned}
$$

Subtracting (57) from (56) to get $\mathcal{I}_{j,1,d'}^{(t+1)}$, the first-order terms $-2\eta(\gamma_j^{\odot})^{(t)}\boldsymbol{\omega}_j^{(t)\top}\boldsymbol{g}_j^{(t)}$ from both expansions cancel, since $(\gamma_j^{\odot})^{(t)} = \gamma_{j,d'}^{(t)} \cdot \prod_{d''\neq d'} \gamma_{j,d''}^{(t)}$, and we obtain after re-grouping of terms:

$$
\mathcal{I}_{j,1,d'}^{(t+1)} = \Big(1 - \frac{4\lambda}{D}\eta\Big)\mathcal{I}_{j,1,d'}^{(t)} + \eta^2\Delta_{j,1,d'}^{(t)}, \quad (58)
$$

where

$$
\begin{aligned}
\Delta_{j,1,d'}^{(t)} &:= \left\|(\gamma_j^{\odot})^{(t)}\boldsymbol{g}_j^{(t)} + \frac{2\lambda}{D}\boldsymbol{\omega}_j^{(t)}\right\|_2^2 - \Big(\prod_{d''\neq d'} \gamma_{j,d''}^{(t)} \cdot \boldsymbol{\omega}_j^{(t)\top}\boldsymbol{g}_j^{(t)} + \frac{2\lambda}{D}\gamma_{j,d'}^{(t)}\Big)^2 \\
&= \|\nabla_{\boldsymbol{\omega}_j}\mathcal{L}(\boldsymbol{\omega}^{(t)}, \boldsymbol{\Gamma}^{(t)})\|_2^2 - \Big(\frac{\partial\mathcal{L}(\boldsymbol{\omega}^{(t)}, \boldsymbol{\Gamma}^{(t)})}{\partial\gamma_{j,d'}}\Big)^2 \quad (59)
\end{aligned}
$$

is the second-order residual that reduces to the imbalance of *gradients*.[2] This further implies that close to stationarity of the $D$-Gated objective, $\Delta_{j,1,d'}^{(t)}$ also vanishes and the discrete-time dynamics reduce to simple geometric decay .

For the scalar-scalar imbalances $\mathcal{I}_{j,d,d'}^{(t+1)}$, we already derived $(\gamma_{j,d'}^{(t+1)})^2$ in Eq. (57), which also symmetrically defines $(\gamma_{j,d}^{(t+1)})^2$. Subtracting both, the first-order terms cancel again, and we obtain

$$
\mathcal{I}_{j,d,d'}^{(t+1)} = \Big(1 - \frac{4\lambda}{D}\eta\Big)\mathcal{I}_{j,d,d'}^{(t)} + \eta^2\big((A_{j,d}^{(t)})^2 - (A_{j,d'}^{(t)})^2\big) := \Big(1 - \frac{4\lambda}{D}\eta\Big)\mathcal{I}_{j,d,d'}^{(t)} + \eta^2\Delta_{j,d,d'}^{(t)}, \quad (60)
$$

where $(A_{j,d}^{(t)})^2 := \big(\prod_{d''\neq d} \gamma_{j,d''}^{(t)} \cdot \boldsymbol{\omega}_j^{(t)\top}\boldsymbol{g}_j^{(t)} + \frac{2\lambda}{D}\gamma_{j,d}^{(t)}\big)^2 = \big(\frac{\partial\mathcal{L}(\boldsymbol{\omega}^{(t)}, \boldsymbol{\Gamma}^{(t)})}{\partial\gamma_{j,d}}\big)^2$. This shows point i).

Similar to the vector-scalar case, $\Delta_{j,d,d'}^{(t)}$ being a difference of squared partial derivatives implies that the term vanishes at stationarity, resulting in geometric decay.

For point ii), we assume balancedness, i.e., $\mathcal{I}_{j,d,d'}^{(t)} = 0$, for any two scalar factors $d, d' > 1$. The imbalance under (S)GD evolves as $\mathcal{I}_{j,d,d'}^{(t+1)} = \big(1 - \frac{4\lambda}{D}\eta\big)\mathcal{I}_{j,d,d'}^{(t)} + \eta^2\big((A_{j,d}^{(t)})^2 - (A_{j,d'}^{(t)})^2\big)$. Therefore, we must show that pair-wise balancedness $\mathcal{I}_{j,d,d'}^{(t)} = 0$ (with potentially non-zero factors $\gamma_{j,d}, \gamma_{j,d'}$) implies $\Delta_{j,d,d'}^{(t+1)} = 0$ for $d, d' > 1$. Abbreviating $\big(\gamma_j^{\odot\backslash d}\big)^{(t)} := \prod_{d''\neq d} \gamma_{j,d''}^{(t)}$, we factor the second-order term as

$$
\Delta_{j,d,d'}^{(t)} = (A_{j,d}^{(t)})^2 - (A_{j,d'}^{(t)})^2 = (A_{j,d}^{(t)} + A_{j,d'}^{(t)})(A_{j,d}^{(t)} - A_{j,d'}^{(t)}). \quad (61)
$$

---

[2]Note that this can be further simplified to $\Delta_{j,1,d'}^{(t)} = \big((\gamma_j^{\odot\backslash d'})^{(t)}\big)^2 \cdot \big((\gamma_{j,d'}^{(t)})^2\|\boldsymbol{g}_j^{(t)}\|_2^2 - (\boldsymbol{\omega}_j^{(t)\top}\boldsymbol{g}_j^{(t)})^2\big) + \frac{4\lambda^2}{D^2}\mathcal{I}_{j,1,d'}^{(t)}$, showing that we can not fully factor $\mathcal{I}_{j,1,d'}^{(t)}$ out of $\Delta_{j,1,d'}^{(t)}$. Hence $\mathcal{I}_{j,1,d'}^{(t)} = 0$ does not generally imply $\mathcal{I}_{j,1,d'}^{(t+1)} = 0$ for vector-scalar imbalances where $d > 1$.

The sum term is $\left((\gamma_j^{\odot\backslash d})^{(t)} + (\gamma_j^{\odot\backslash d'})^{(t)}\right) \cdot \boldsymbol{\omega}_j^{(t)\top} \boldsymbol{g}_j^{(t)} + \frac{2\lambda}{D}(\gamma_{j,d}^{(t)} + \gamma_{j,d'}^{(t)})$, and the difference term $\left((\gamma_j^{\odot\backslash d})^{(t)} - (\gamma_j^{\odot\backslash d'})^{(t)}\right) \cdot \boldsymbol{\omega}_j^{(t)\top} \boldsymbol{g}_j^{(t)} + \frac{2\lambda}{D}(\gamma_{j,d}^{(t)} - \gamma_{j,d'}^{(t)})$. Expanding their product, we obtain:

$$\Delta_{j,d,d'}^{(t)} = \left((\gamma_j^{\odot\backslash d})^{(t)} + (\gamma_j^{\odot\backslash d'})^{(t)}\right)\left((\gamma_j^{\odot\backslash d})^{(t)} - (\gamma_j^{\odot\backslash d'})^{(t)}\right)(\boldsymbol{\omega}_j^{(t)\top}\boldsymbol{g}_j^{(t)})^2 + \left(\frac{4\lambda^2}{D^2}\right)\left((\gamma_{j,d}^{(t)})^2 - (\gamma_{j,d'}^{(t)})^2\right)$$
$$+ \left(\frac{2\lambda}{D}\right)\left(\gamma_{j,d}^{(t)} + \gamma_{j,d'}^{(t)}\right)\left((\gamma_j^{\odot\backslash d})^{(t)} - (\gamma_j^{\odot\backslash d'})^{(t)}\right)\boldsymbol{\omega}_j^{(t)\top}\boldsymbol{g}_j^{(t)}$$
$$+ \left(\frac{2\lambda}{D}\right)\left(\gamma_{j,d}^{(t)} - \gamma_{j,d'}^{(t)}\right)\left((\gamma_j^{\odot\backslash d})^{(t)} + (\gamma_j^{\odot\backslash d'})^{(t)}\right)\boldsymbol{\omega}_j^{(t)\top}\boldsymbol{g}_j^{(t)}. \tag{62}$$

The last two mixed terms cancel, which can be seen by summing them and factoring:

$$\left(\frac{2\lambda}{D}\right)\boldsymbol{\omega}_j^{(t)\top}\boldsymbol{g}_j^{(t)}\cdot\Big[\left(\gamma_{j,d}^{(t)} + \gamma_{j,d'}^{(t)}\right)\left((\gamma_j^{\odot\backslash d})^{(t)} - (\gamma_j^{\odot\backslash d'})^{(t)}\right)$$
$$+ \left(\gamma_{j,d}^{(t)} - \gamma_{j,d'}^{(t)}\right)\left((\gamma_j^{\odot\backslash d})^{(t)} + (\gamma_j^{\odot\backslash d'})^{(t)}\right)\Big] \tag{63}$$

Now, expanding the bracket, all terms but $2\gamma_{j,d}^{(t)}(\gamma_j^{\odot\backslash d})^{(t)} - 2\gamma_{j,d'}^{(t)}(\gamma_j^{\odot\backslash d'})^{(t)} = 2(\gamma_j^{\odot})^{(t)} - 2(\gamma_j^{\odot})^{(t)} = 0$ cancel, so that the remaining expression for the second-order residual is

$$\Delta_{j,d,d'}^{(t)} = \left(((\gamma_j^{\odot\backslash d})^{(t)})^2 - ((\gamma_j^{\odot\backslash d'})^{(t)})^2\right)\left(\boldsymbol{\omega}_j^{(t)\top}\boldsymbol{g}_j^{(t)}\right)^2 + \left(\frac{4\lambda^2}{D^2}\right)\mathcal{I}_{j,d,d'}^{(t)}. \tag{64}$$

Finally, using $(\gamma_j^{\odot\backslash d})^{(t)} = \gamma_{j,d'}^{(t)} \cdot (\gamma_j^{\odot\backslash d,d'})^{(t)} := \gamma_{j,d'}^{(t)} \cdot \prod_{d''\in[D-1]\backslash\{d,d'\}}\gamma_{j,d''}^{(t)}$, we can simplify

$$\Delta_{j,d,d'}^{(t)} = \left(\left((\gamma_j^{\odot\backslash d})^{(t)}\right)^2 - \left((\gamma_j^{\odot\backslash d'})^{(t)}\right)^2\right)(\boldsymbol{\omega}_j^{(t)\top}\boldsymbol{g}_j^{(t)})^2 + \left(\frac{4\lambda^2}{D^2}\right)\mathcal{I}_{j,d,d'}^{(t)}$$
$$= ((\gamma_{j,d'}^{(t)})^2 - (\gamma_{j,d}^{(t)})^2)((\gamma_j^{\odot\backslash d,d'})^{(t)})^2(\boldsymbol{\omega}_j^{(t)\top}\boldsymbol{g}_j^{(t)})^2 + \left(\frac{4\lambda^2}{D^2}\right)\mathcal{I}_{j,d,d'}^{(t)}$$
$$= \mathcal{I}_{j,d,d'}^{(t)}\Big[-(\boldsymbol{\omega}_j^{(t)\top}\boldsymbol{g}_j^{(t)})^2 \cdot ((\gamma_j^{\odot\backslash d,d'})^{(t)})^2 + \frac{4\lambda^2}{D^2}\Big] \tag{65}$$

This shows for scalar gates $\gamma_{j,d}, \gamma_{j,d'}$ with $d, d' \in [D]\backslash\{1\}$, all terms in $\mathcal{I}_{j,d,d'}^{(t+1)}$ depend multiplicatively on $\mathcal{I}_{j,d,d'}^{(t)}$, and hence $\mathcal{I}_{j,d,d'}^{(t)} = 0$ implies $\mathcal{I}_{j,d,d'}^{(t+1)} = (1 - \frac{4\lambda}{D}) \cdot 0 + \eta^2 \cdot 0 = 0$, proving point ii).

For point iii), note that a balanced zero representation $(\boldsymbol{\omega}_j^{(t)}, \{\gamma_{j,d}^{(t)}\}_{d=1}^{D-1}) = \boldsymbol{0}$ of all vector-scalar and scalar-scalar factor pairs satisfies $\mathcal{I}_{j,d,d'}^{(t)} = 0$ for all $d, d' \in [D]$. Since all terms in $\Delta_{j,d,d'}^{(t+1)}$ are multiplicative in either $\boldsymbol{\omega}_j^{(t)}$ or $\gamma_{j,d}^{(t)}$, it vanishes. Therefore, $\mathcal{I}_{j,d,d'}^{(t+1)} = (1 - \frac{4\lambda}{D}\eta) \cdot 0 + \eta^2 \cdot 0 = 0$ for $d, d' \neq 1$. Analogously, $\Delta_{j,1,d'}^{(t+1)} = 0$ using the same argument. Hence $\mathcal{I}_{j,1,d'}^{(t+1)} = 0$. Beyond simply enforcing balancedness between both vector and scalar gating factors, inspecting the updates for $\boldsymbol{\omega}_j^{(t)} = \boldsymbol{0}$ and $\gamma_{j,d}^{(t)} = 0$ in Eq. (55) directly shows that $\boldsymbol{\omega}_j^{(t+1)} = \boldsymbol{0} - \eta \cdot \boldsymbol{0} = \boldsymbol{0}$ and $\gamma_{j,d}^{(t+1)} = 0 - \eta \cdot 0 = 0$. By induction, the iterates of a balanced zero representation in group $j \in [J]$ are thus confined to $\mathbf{w}_j^{(t')} = \boldsymbol{0} \,\forall\, t' > t$ under (S)GD dynamics. This completes the proof. □

## C    Experimental details

In all our experiments, we evaluate $D$-Gating only for $D \in \{2, 3, 4\}$, resulting in induced $L_{2,1}, L_{2,2/3}$, and $L_{2,0.5}$ regularization, following broadly established ranges for $L_{p,q}$ group regularization [22]. Although an ever deeper $D$-Gating parametrization better approximates a group $L_0$ penalty, the increase in non-convexity potentially causes numerical instabilities without necessarily improving performance beyond 4-Gating (cf. Appendix E.3). Nevertheless, in the majority of our experiments, there is a marked distinction between convex group penalization ($D = 2$) and the non-convex extension ($D > 2$). Throughout all experiments, $\boldsymbol{\omega}$ is initialized using a standard initialization scheme as for w, while the gating parameters $\gamma_{j,d}$ are initialized with unity, i.e., $\boldsymbol{\Gamma}^{(0)} = \mathbf{1}_J \mathbf{1}_{D-1}^\top$, see Algorithm 1.

Table 2: Summary of datasets used in feature selection experiments.

| Dataset | Training Samples | Test Samples | Classes | Input Features |
|---|---|---|---|---|
| ISOLET [9] | 6,328 | 1,559 | 26 | 617 |
| COIL20 [44] | 1,152 | 288 | 20 | 400 |
| ACTIVITY [2] | 4,252 | 1,492 | 6 | 561 |
| MNIST [34] | 60,000 | 10,000 | 10 | 784 |
| F-MNIST [64] | 60,000 | 10,000 | 10 | 784 |
| MADELON [17] | 2,080 | 520 | 2 | 500 |

## C.1 Details on group lasso simulation

**Simulation details** For our group sparse linear regression set-up, we simulate $n = 200$ training (and 2000 test) samples with $p = 200$ features grouped into 40 groups of 5 features each. 7 feature groups contain informative (non-zero) weights and 33 groups contain noise features only. We draw both the feature values and the (informative) weights from $\mathcal{N}(0, 1)$ and likewise add standard Gaussian additive noise to the simulated predictor to obtain the targets.

**Training details** We train the $D$-Gated models and direct $L_{2,1}$ penalization using full-batch gradient descent for 1500 iterations using a cosine learning rate schedule with initial learning rate $5 \times 10^{-2}$ and 0.9 momentum. We consider parameter groups with norm smaller than $10^{-6}$ to be zero. For the specialized optimization routine to solve the Group Lasso efficiently, we use the accelerated proximal gradient method proposed by [50]. The *oracle* method shown in Fig. 4 is the efficient ordinary least squares estimator computed only on the truly non-zero coefficients. Within each simulation, all regularized methods are run on a grid of 30 $\lambda$ values spaced logarithmically between $10^{-5}$ and 15. The procedure is repeated over 10 simulations and the average results reported in Fig. 4.

## C.2 Details on feature selection experiments

**Architecture** As a backbone for all methods ($D$-Gating, LassoNet, HSIC), we use a fully-connected LeNet-300-100 architecture [35] with two hidden layers (300 and 100 neurons) and ReLU activation functions and Kaiming Normal initialization. To obtain feature selection for non-linear models using our approach, we group the first-layer weights of the network by input feature and apply $D$-Gating. We train the network on all classification tasks and methods using SGD for 100 epochs with a batch size of 256 and cosine schedule with an initial learning rate of 0.1. All models are initialized using a Kaiming Normal scheme, with the scaling factors for $D$-Gating being initialized with unity (cf. Algorithm 1). Moreover, we consider parameter groups with 2-norm smaller than `float32.mach.eps` $\approx 1.19 \times 10^{-7}$ to be 0. Table 2 describes the basic properties of the used datasets. For datasets which do not explicitly provide a test set, we randomly assign 20% of the samples to the test set. Note that, to ensure a fair comparison to $D$-Gating, LassoNet [37] is implemented without an additional debiasing step that retrains the model only on the selected features.

## C.3 Details on filter sparsity experiments

**Architectures** VGG-16 for CIFAR-10 and SVHN consists of 13 convolutional layers and 3 fully-connected layers [51]. The convolutional layers are organized into five groups: two layers with 64 filters, two with 128 filters, 3 layers with 256 filters, 3 layers with 512 filters, and another 3 layers with 512 filters. After each group, we apply a $2 \times 2$ max-pooling operation. All filters are $3 \times 3$. We add batch normalization immediately before every ReLU activation. To reduce parameters compared to the ImageNet version, we follow [68] and replace the two dense layers before the output with a single fully-connected layer of 512 neurons.
ResNet-18 is an 18-layer deep residual network introduced by [19]. For small-image datasets, we follow, e.g., [54], by replacing the initial $7 \times 7$ convolution and the following max-pooling layer with a single $3 \times 3$. The model begins with that convolution, then proceeds through four stages of basic residual blocks, each stage consisting of two blocks, with filter counts of $[64, 128, 256, 512]$. Finally, global average pooling reduces the feature maps before the fully connected output layer. In all methods we only regularize the convolutional but not the fully-connected layers, either through

Table 3: Summary of datasets used in image classification experiments.

| Dataset | Training Samples | Test Samples | Classes | Input Features |
|---|---|---|---|---|
| CIFAR-10 [31] | 50,000 | 10,000 | 10 | 3,072 ($32\times32\times3$) |
| SVHN [45] | 73,257 | 26,032 | 10 | 3,072 ($32\times32\times3$) |
| CIFAR-100 [31] | 50,000 | 10,000 | 100 | 3,072 ($32\times32\times3$) |

$D$-Gating or by directly regularizing the filter weights using an $L_{2,1}$ penalty [62, 49, 38] or the batch normalization scaling variables as in network slimming [40].

**Data** We conduct experiments on three standard image classification benchmark tasks described in Table 3. We use the train/test split provided by the datasets. Additionally, we use standard pre-processing and data augmentation techniques, namely normalizing the input images and employing horizontal flips, width or height shifts up to $12.5\%$, and up to $15°$ rotations.

**Training** Our training hyperparameter settings generally follow broadly established configurations for the respective tasks [51, 19, 68]. Because established learning rate settings were found to perform suboptimally, we scanned the interval $[10^{-3}, 1]$ using a small regularization strength $\lambda$ for each method and dataset separately. Table 4 lists the hyperparameters for the filter sparsity experiments.

Table 4: Training hyperparameters for different architectures and image classification datasets. The learning rates correspond to $D$-Gating with $D = \{2, 3, 4\}$ (set) and the comparison methods (second value). The comparison methods use standard Kaiming initialization.

| Architecture | Dataset | Epochs | Batch size | Optim. | Mom. | Init. | LR | Schedule |
|---|---|---|---|---|---|---|---|---|
| **VGG-16** | CIFAR-10 | 200 | 128 | SGD | 0.9 | Kaiming Normal | {0.3,0.4,0.4}, 0.1 | Cosine |
| | SVHN | 200 | 128 | SGD | 0.9 | Kaiming Normal | {2e-3, 3e-3, 3e-3}, 2e-3 | Cosine |
| **ResNet-18** | CIFAR-100 | 200 | 128 | SGD | 0.9 | Kaiming Normal | {0.2,0.3,0.4}, 0.2 | Cosine |

In $D$-Gating, the sparsity is controlled by the regularization strength $\lambda$. To obtain the accuracy-sparsity tradeoffs, we therefore train models for each $D \in \{2, 3, 4\}$ along a grid of $\lambda$ values that spans the whole sparsity range. For the comparison methods, direct optimization of the sparsity-inducing $L_{2,1}$ and $L_1$ penalties does not result in sparse filters whose norm is below `float32.mach.eps` $\approx 1.19 \times 10^{-7}$. To achieve sparsity and construct the tradeoff curves, we also train these models along a grid of $\lambda$ values but subsequently post-hoc prune each of these models along a sequence of pre-defined pruning ratios $\{0.1, 0.2, \ldots, 0.9, 0.95, 0.98, 0.99\}$. The tradeoff curves are then constructed by taking the best performance over all $\lambda$ values separately at each pruning ratio. This optimally reflects different regularization requirements at different sparsity ranges. In contrast to the original proposals of [62, 38, 40], we implement both direct $L_{2,1}$ regularization with post-hoc filter pruning and network slimming ($L_1$ regularization of the batch normalization scaling parameters) as a one-shot procedure without retraining after sparsification to ensure a fair comparison to $D$-Gating, which can be seen as merely another optimization vehicle to optimize non-smooth structured sparsity penalties, making them comparable outside of a sophisticated pruning pipeline. Moreover, like direct optimization of these penalties, $D$-Gating can be arbitrarily combined with other pruning and retraining schemes.

### C.4 Details on language modeling experiments

We choose a variant of NanoGPT trained on the character-level TinyShakespeare dataset comprising $5,458,199$ tokens and a vocabulary size of 91 different symbols. Deviating from the usual performance evaluation metrics for language models, the character-level granularity also allows for reporting the validation accuracy instead of only the perplexity. We set aside $10\%$ of the training data for validation purposes.

Our NanoGPT implementation contains 10.8 mio. parameters and has 6 layers, 12 attention heads per multi-head attention layer, block size of 256, an embedding dimension of 384 and dropout with parameter 0.2 between the layers. We train all models on a grid of logarithmically spaced $\lambda$ values between $10^{-4}$ and $10^2$ for 3000 iterations using a batch size of 32 and Adam [26] with a cosine schedule and initial learning rate $10^{-3}$. Since direct optimization of the $L_{2,1}$ penalty does

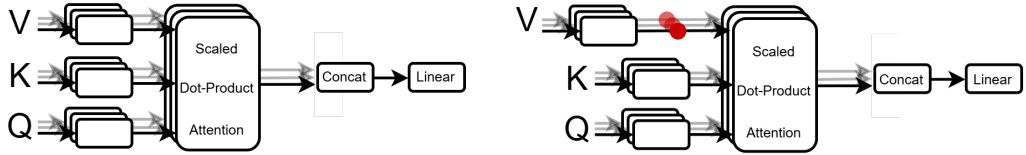

Figure 9: Schematic illustration of $D$-Gating overparameterization to achieve structured head sparsity in a multi-head attention layer. **Left**: original attention layer. **Right**: $D$-Gated value matrices sparsify the whole attention head. Red nodes indicate the location of the additional gating parameters.

not yield sparse solutions, defined as having a 2-norm smaller than $10^{-5}$, we further prune these models using structured global magnitude pruning to sparsity ratios $\{0.05, 0.1, 0.2, \ldots, 0.9, 0.95\}$, and report the best performance over all $\lambda$ values for a given sparsity ratio. Note that the $D$-Gated models incorporate no additional pruning.

Fig. 9 illustrates how we implement structured attention head penalization using $D$-Gating. Note that we only gate the value matrices of each head, which suffices for the whole attention head to be removed when sparsified.

### C.5 Details on Neural Oblivious Decision Ensembles

We optimize the squared loss on the Wine quality prediction task ($n = 4{,}898; p = 11$) [7] using SGD with Nesterov momentum $0.9$ and a cosine decay learning rate schedule with initial learning rate $0.4$. The models are trained for $400$ epochs without early stopping using a batch size of $1024$. Fig. 10 describes the application of $D$-Gating in NODEs for $D = 2$.

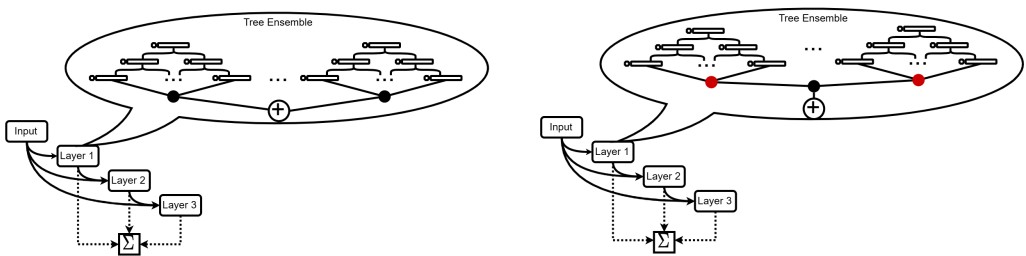

Figure 10: Schematic illustration of $D$-Gating overparameterization to achieve structured tree sparsity in a NODE model. **Left**: original NODE structure. **Right**: NODE with 2-gated tree structures (multiplied by a shared factor).

## D  Computational Environment

All experiments were conducted either on a single NVIDIA RTX A6000 or A4000 GPU with 48GB and 16GB of memory, respectively. Smaller experiments, e.g., for toy models, were carried out on standard CPU workstations. The total single-GPU runtime for all experiments is estimated to be around $500$ hours.

## E  Further experiments and ablation studies

### E.1  Additional plots for filter sparsity in image classification

Besides the accuracy-sparsity tradeoff, the reduction of FLOPs through is of special interest for structured sparsification methods. Besides Table 1, we show the full tradeoff curves for accuracy and theoretical speed-up, measured as the ratio of FLOPs of the original and sparse models, in Fig. 11.

Additionally, Fig. 12 shows the regularization paths for $D$-Gating with $D \in \{2, 3, 4\}$ for all three image classification datasets we experimented on. Complementing the conclusions from the tradeoff

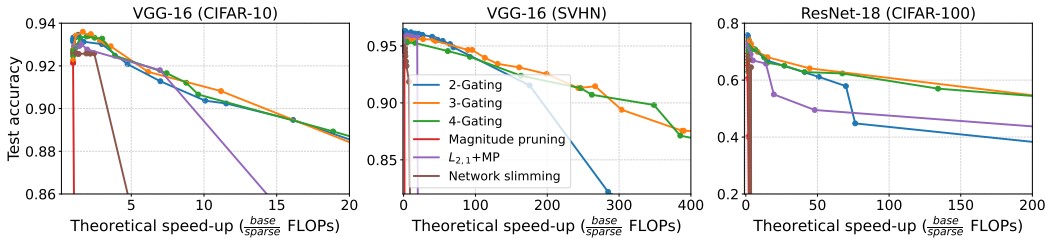

Figure 11: Test accuracy vs. theoretical speed-up (measured as the ratio of FLOPs) of $D$-Gated convolutional neural networks and comparison methods.

curves in the main text, both plots corroborate our findings of superior tradeoffs for $D = \{3, 4\}$ over $D = 2$, as well as increased sparsity at the same regularization parameter $\lambda$.

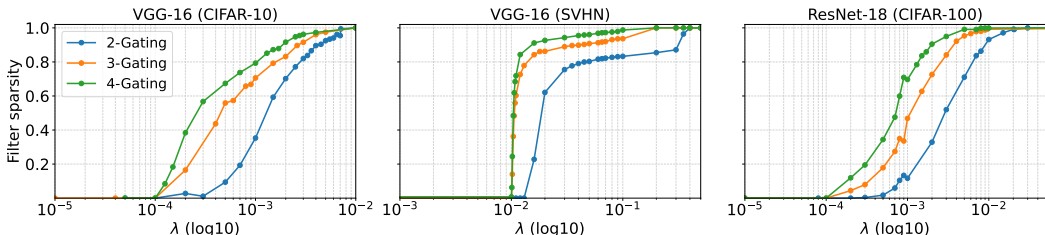

Figure 12: Effect of regularization $\lambda$ on filter sparsity for $D \in \{2, 3, 4\}$.

## E.2 Overheads

**Parameter overhead** To quantify the parameter overhead incurred by applying $D$-Gating in our experiments, we contrast the number of additional parameters with the models' parameters. The results are shown in Table 5, highlighting that larger architectures typically have fewer regularized groups (attention heads, convolutional filters) with large group sizes of thousands of parameters, suggesting that our approach scales better for larger than for small architectures by introducing relatively fewer additional parameters.

Table 5: Parameter count overhead for $D$-Gating. The linear model is the one from Fig. 4, the input-sparse LeNet-300-100, the one from Fig. 5, VGG-16 from Fig. 6, and NanoGPT from Fig. 7. The overhead rate in the worst case ($D = 4$ and the largest input dimension), is given by $\frac{\text{\#new params}}{\text{\#old params}} - 1$.

| Model | Vanilla Params | Additional Params | Overhead rate ($D = 4$) | Details |
|---|---|---|---|---|
| Lin. Mod. | 200 | $40 \cdot (D-1)$ | $6.0 \times 10^{-1}$ | 40 groups of 5 features |
| LeNet-300-100 | $\approx 266{,}000$ | $\texttt{input\_dim} \cdot (D-1)$ | $8.8 \times 10^{-3}$ | $\texttt{input\_dim} \in [400, 784]$ |
| VGG-16 | $\approx 15$ mio. | $4{,}224 \cdot (D-1)$ | $8.4 \times 10^{-4}$ | Sum of conv. filters in 13 layers |
| NanoGPT | $\approx 10.8$ mio. | $72 \cdot (D-1)$ | $2.0 \times 10^{-5}$ | 6 layers, 12 att. heads |

**Runtime and memory overhead** Additionally, we measure the wall-clock runtime overhead of $D$-Gating for $D \in \{2, 3, 4\}$ against the vanilla models. Similarly, we record the peak GPU memory utilization during training.

The runtime results are shown in Fig. 13, showing slight to moderate increases between $0\%$ and $30\%$ for smaller batch sizes, with the difference becoming increasingly irrelevant for commonly used larger batch sizes as well as larger architectures.

Similarly, Fig. 14 shows the memory overhead over vanilla models, indicating indiscernible utilization for the vast majority of settings. The behavior of the fully-connected LeNet-300-100 exhibits high variance and less clear patterns compared to the larger models, potentially due to its small size. Generally, part of the difference in runtime and memory overhead is likely due to implementation efficiency reasons and not necessarily an inherent difference.

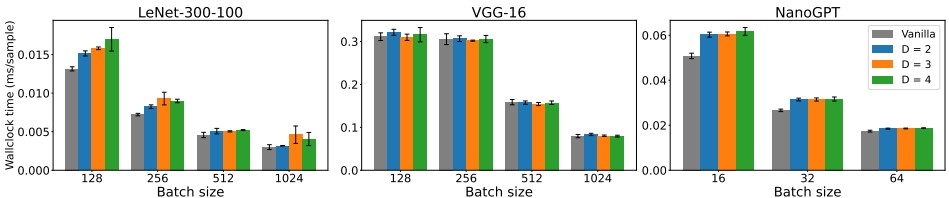

Figure 13: Wall-clock runtime for $D \in \{2, 3, 4\}$ and vanilla model. Means and std. over ten runs are shown.

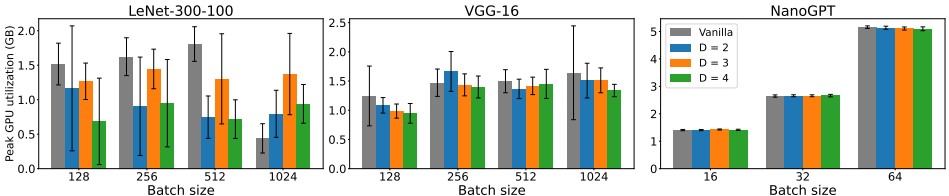

Figure 14: Peak memory use for $D \in \{2, 3, 4\}$ and vanilla model. Means and std. over ten runs are shown.

### E.3 Depth and instability ablation studies

To guide the selection of the gating depth $D$ and demonstrate that the full benefits of non-convex $L_{2,q}$, $q < 1$, regularization are usually attained at $q = 2/3$ or $q = 0.5$, corresponding to $D = 3, 4$ for $D$-Gating [22], we further conduct depth ablation studies for the input-sparse LeNet-300-100 from the experiment in Fig. 5 on ISOLET, extend the results for the filter-sparse VGG-16 on CIFAR-10 (Fig. 6), and analyze performance and instability for the group sparse regression task in Fig. 4.

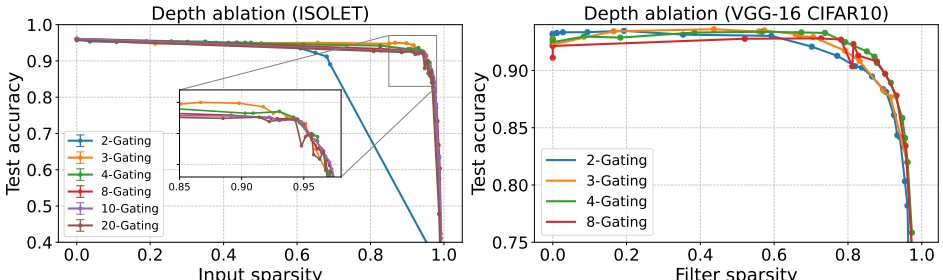

Figure 15: Effect of increasing the gating depth $D$ beyond $D = 4$ for the input-sparse LeNet-300-100 on ISOLET as well as the filter-sparse VGG-16 trained on CIFAR-10.

Fig. 15 shows the results for LeNet-300-100 and VGG-16 for increasing depth up to $D = 20$ and $D = 8$, respectively. Importantly, we observe no measurable improvement in the performance-sparsity tradeoff beyond $D = 4$, while the instability increases slightly.

For the group sparse linear regression task shown in Fig. 16, the left panel illustrates the numerical instability caused by too large $D$, evidenced by the jagged regularization paths for $D > 6$. The tradeoff curves in the right panel reveal that these instabilities degrade the performance at lower sparsity values for large $D$.

Taken together, the values of $q \in \{2/3, 0.5\}$, i.e., $D \in \{3, 4\}$, recommended in the literature are supported by our ablation studies, where deeper gating approaches show at best equivalent performance while increasing numerical instability.

### E.4 Imbalance decay and loss convergence for SGD and Adam

The convergence of the $D$-Gating objective $\mathcal{L}(\boldsymbol{\omega}, \boldsymbol{\Gamma})$ to the non-smooth regularized objective $\mathcal{L}_{\mathbf{w}}(\mathbf{w})$, as shown in Fig. 3, is directly related to the convergence of the regularizers of both objectives, as well

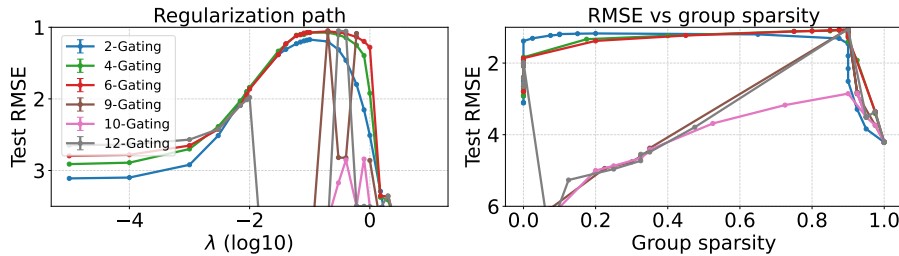

Figure 16: Effect of increasing the gating depth $D$ beyond $D = 4$ on the group-sparse linear regression task. The left panel shows the regularization paths, and the right panel the performance-sparsity tradeoffs. The setup is the same as in Fig. 4.

as the balancedness of the gating representations. To see this, note that

$$
\begin{aligned}
\mathcal{L}(\boldsymbol{\omega}, \boldsymbol{\Gamma}) - \mathcal{L}_{\mathbf{w}}(\mathbf{w}) &= \mathcal{L}_0(\boldsymbol{\omega} \blacktriangleright \boldsymbol{\gamma}^{\odot}) + \lambda(\|\boldsymbol{\omega}\|_2^2 + \|\boldsymbol{\Gamma}\|_F^2)/D - \mathcal{L}_0(\mathbf{w}) - \lambda\|\mathbf{w}\|_{2,2/D}^{2/D} \\
&= \lambda((\|\boldsymbol{\omega}\|_2^2 + \|\boldsymbol{\Gamma}\|_F^2)/D - \|\mathbf{w}\|_{2,2/D}^{2/D}) \\
&= \lambda(\mathcal{R}(\boldsymbol{\omega}, \boldsymbol{\Gamma}) - \mathcal{R}_{\mathbf{w}}(\mathbf{w})) \\
&= \lambda\mathcal{M}(\boldsymbol{\omega}, \boldsymbol{\Gamma}),
\end{aligned}
\tag{66}
$$

Hence, the losses converge if the regularizers converge, which, by the AM-GM inequality, happens if and only if the gating representations are balanced. This relationship is verified in Fig. 17, showing the same experiment as in Fig. 3, but plotting the convergence of regularizers instead of losses.

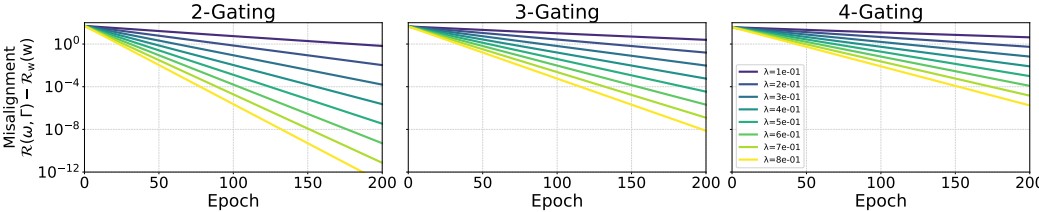

Figure 17: Convergence of regularizers, i.e., evolution of misalignment, instead of loss convergence for SGD trained on a neuron-sparse LeNet-300-100. Setting is the same as for Fig. 3.

Another interesting question is to empirically investigate the evolution of misalignment for a common optimizer, for which the gradient flow and descent results Lemma 3 and Lemma 4 do not apply, i.e., Adam [26]. Fig. 18 shows the evolution of misalignment for the same setting as the previous Fig. 17, but optimized with Adam instead of SGD. Results show that the $D$-Gating objective also converges to the sparsity-inducing objective $\mathcal{L}_{\mathbf{w}}(\mathbf{w})$ faster than SGD overall, albeit not at a neat exponential rate. Moreover, the convergence speeds do not seem to depend as strongly on $D$ as for SGD.

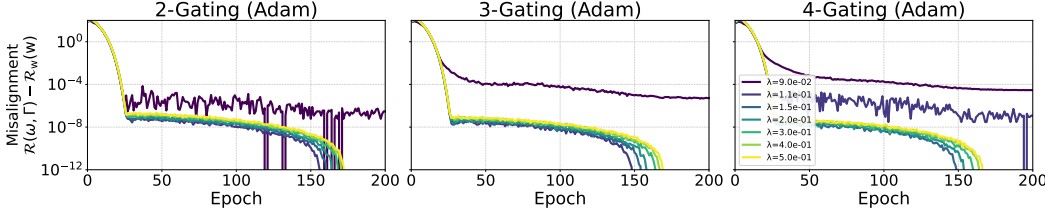

Figure 18: Convergence of regularizers with Adam instead of SGD trained on a neuron-sparse LeNet-300-100. Setting is the same as for Fig. 3.

### E.5 Subnetwork selection

In an additional application, we investigate whether $D$-Gating can correctly select whole data modalities in a simple late-fusion multimodal network [13]. In these architectures, the modalities are processed independently in separate subnetworks before their latent representations are fused in the penultimate late-fusion layer. Fig. 19 visualizes such an architecture as well as our $D$-Gating proposal for differentiable selection of modalities. In the $D$-Gating variant, the final-layer weights are grouped by modality and subsequently gated, effectively selecting out whole modality subnetworks. To control the contribution of the modalities, we simulate a semi-synthetic dataset combining both

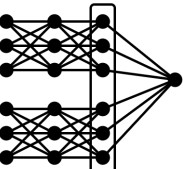 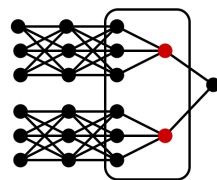

Figure 19: Schematic illustration of modality selection in a late-fusion multimodal network using $D$-Gating overparameterization. **Left**: original late-fusion architecture. **Right**: architecture including $D$-Gating of the late-fusion layer, where the red nodes indicate the location of the gating factors.

tabular and image information additively in a predictor. To be precise, we simulate targets as

$$Y = (1 - \alpha) \cdot \eta^{\text{tabular}} + \alpha \cdot \eta^{\text{image}} + \varepsilon, \quad \varepsilon \sim \mathcal{N}(0, 1), \, \alpha \in \{0, 0.5, 1\}.$$

The image predictor is based on the MNIST dataset of handwritten images and is simply the (demeaned) face value of the digit contained in the image. The tabular predictor is given by $\eta^{\text{tabuar}} = \sum_{k=1}^{6} f_k(x_k)$, $x_k \sim \mathcal{U}[-1, 1]$, where the $f_k(\cdot)$ are fixed smooth non-linear shape functions. Both predictors are constructed to be similarly distributed. We simulate $n = 2000$ training and test observations from the above data-generating process. In the late-fusion architecture, we use a small VGG-style subnetwork with 4 convolutional layers for the MNIST images and a shallow subnetwork to model the non-linear effects $f_k(\cdot)$ of the tabular covariates $x_k$. The models are optimized for 300 epochs using SGD with $0.8$ momentum, a batch size of 32, and a learning rate of $5 \times 10^{-3}$. Fig. 20 shows the results for $D \in \{2, 3, 4\}$ and $\alpha \in \{0, 0.5, 1\}$. As expected, using the one standard error rule to select the regularization strength $\lambda$, $D$-Gating can consistently select out the unimportant modalities, with the separation of important and unimportant modalities becoming clearer as $D$ increases. Notably, for equal modality contributions (middle row), neither the validation-optimal $\lambda_{opt}$ nor $\lambda_{1se}$ selects out any of the modalities.

### E.6 Differentiable sparse neural additive models

In another application, we show that $D$-Gating can be seamlessly applied to achieve differentiable shape function sparsity in Neural Additive Models (NAMs) [1], which we call $D$-SNAMs. In NAMs, each feature is processed independently in a separate subnetwork to learn a flexible shape function non-parametrically. Then, the learned functions are combined additively to obtain the final inherently interpretable predictor. Fig. 21 visualizes such an architecture as well as our $D$-Gating proposal for differentiable shape function selection in NAMs. In the $D$-Gating variant, all outgoing first-layer weights from each feature are grouped and subsequently gated, effectively selecting out shape functions altogether. A related approach we compare against, Sparse Neural Additive Models (SNAMs), imposes a non-differentiable $L_{2,1}$ group lasso penalty on the shape function weights and proceeds by direct optimization using SGD, i.e., its subgradient variant [65]. We further compare against direct optimization using SGD with an $L_1$ or $L_2$ penalty on the same weights.

We investigate the behavior of $D$-SNAM, i.e., $D$-Gating in NAMs, on synthetic data simulated as follows. The sparse non-linear additive data-generating process (DGP) for the response is given by

$$Y = \sin(x_1) + \cos(x_2) + x_3^2 - x_4 - |x_5| + \sum_{j=6}^{20} 0 \cdot x_j + \varepsilon, \quad \varepsilon \sim \mathcal{N}(0, 1),$$

with five informative features $x_1, \ldots, x_5$ and additional noise covariates $x_6, \ldots, x_{20}$, all drawn independently from $\mathcal{N}(0, 1)$. Thus, only the first five features contribute a non-zero signal, while the

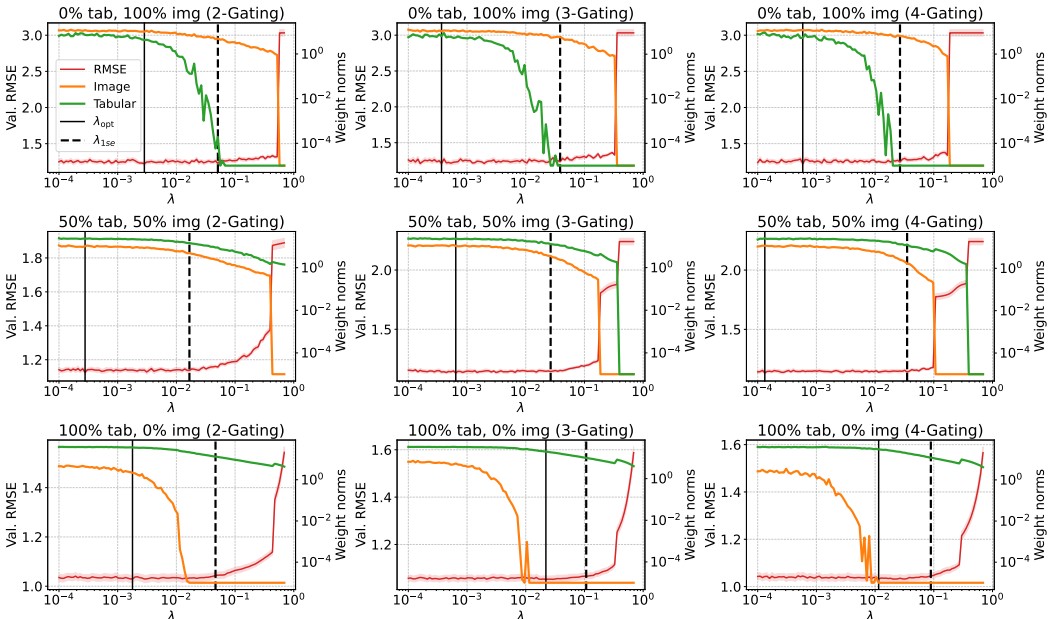

Figure 20: Regularization path and weight norms in multi-modal late-fusion network for different modality contributions (**rows**) and $D$-Gating depth $D \in \{2, 3, 4\}$ (**columns**). Norms below $10^{-5}$ are clipped. The solid vertical bar indicates the validation-optimal $\lambda_{opt}$ and the dashed vertical bar the $\lambda_{1se}$ chosen by the 1se rule, i.e., the sparsest model that performs within one standard error of the lowest validation RMSE (10 splits).

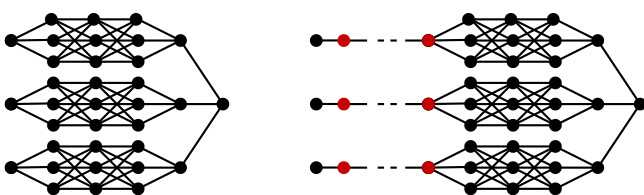

Figure 21: Schematic illustration of differentiable shape function selection in Neural Additive Models using $D$-Gating overparameterization. **Left**: original NAM architecture. **Right**: differentiable sparse $D$-SNAM architecture including $D$-Gating for each feature, where the red nodes indicate the locations of the gating factors.

remaining 15 are uninformative noise. We draw $n_{\text{total}} = 1000$ independent samples from the DGP, and perform 5-fold cross-validation (CV) to obtain mean performance and sparsity metrics, together with standard errors. The regularization strength $\lambda$ varies on a grid between $10^{-2}$ and $10^{1}$.

The NAMs are defined to have identical shape function subnetworks with layers of size $(100, 100, 64, 1)$, respectively, ReLU activations for the hidden layers, and batch normalization [24] after each hidden layer. Optimization of the squared loss is performed for 1000 epochs using Adam optimizer with the default learning rate $10^{-3}$ and batch size 32 without early stopping. Shape functions with a weight norm smaller than $10^{-3}$ are considered zero.

Fig. 22 shows the results for $D \in \{2, 3, 4\}$ as well as the comparison methods SNAM and direct $L_1$ and $L_2$ regularization. Across settings, $D$-Gating achieves differentiable shape function sparsity while SNAM is not able to shrink the weights sufficiently and demands a post-training pruning step. Further, $D$-SNAM effectively removes irrelevant shape functions while retaining informative feature effects. For non-convex induced regularization using $D > 2$, the CV-optimal models are much sparser than both $D = 2$ and the dense SNAM results. Moreover, compared to the direct optimization used in the baselines, $D$-Gating with differentiable group sparsity achieves a markedly lower CV error for $D \in \{3, 4\}$ (left plot in Fig. 22).

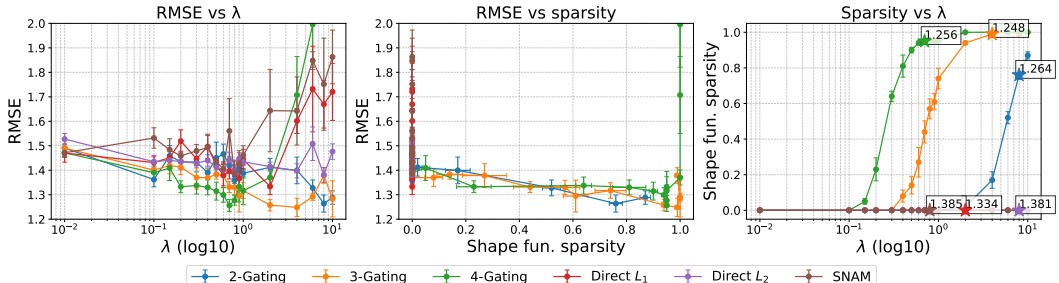

Figure 22: Performance of sparse NAMs with differentiable $D$-Gating for $D \in \{2, 3, 4\}$, compared against $L_1$, $L_2$, and $L_{2,1}$ (SNAM) baselines. Means and standard errors for 5-fold cross-validation are reported. The annotated stars in the right plot indicate the CV-optimal models for each method and their CV error.

