# OpenReview forum: "Differentiable Sparsity via $D$-Gating: Simple and Versatile Structured Penalization"
_NeurIPS.cc/2025/Conference — NeurIPS 2025 spotlight_

### Official Review · Reviewer_4J3D · 2025-06-15

**Clarity:** 3
**Significance:** 2
**Originality:** 3
**Rating:** 4
**Confidence:** 3

**Summary:**

Group sparsity (the removal of entire filters, attention heads, and so on) is a hardware-friendly alternative to unstructured sparsity. This paper proposes a method called D-Gating, which induces group sparsity through a differentiable reparameterization. Instead of applying a non-differentiable L2-over-D group norm penalty directly, the authors introduce a smooth surrogate by multiplying each of each group’s weights by D differentiable (group-specific) gating parameters. This leads to an overparameterized but fully differentiable objective that encourages sparsity in a way equivalent to the original L2-over-D penalty. After training, the gates can be collapsed into the primary parameters, and groups with zeroed gates can be pruned. Theoretical results are developed for SGD, and small-scale experiments show more stable sparsity induction compared to magnitude pruning or direct optimization of the non-differentiable objective.

**Questions:**

Aside from any clarifications from the criticisms above, I'm just curious about the title: what is the key distinction between "modular" and "versatile" that is being implied here?

**Ethical Concerns:**

["NO or VERY MINOR ethics concerns only"]

**Final Justification:**

As noted below: I have read the other reviews and rebuttals, and the rebuttal to my own review.  I feel the responses help improve the clarity and quality of the work.  However, I believe we are still a few steps away from this method being used at large scales, e.g., for LLM pre-training.  Not least because the experiments in this paper are at such a small scale.  The authors say that their contribution is methodological, and the results serve only to "showcase" or "demonstrate" that it works, but as far as I can tell, there's nothing here to necessarily demonstrate that this method will provide consequential advantages at very large scales.

**Limitations:**

The paper could have benefited from a separate limitations section that really outlined the theoretical assumptions that may be violated in practice, e.g., in the very experiments used in the paper!  Moreover, as noted above, I felt the discussion of computational overheads was inadequate, and it should be clearly quantified, rather than simply saying "negligible" or not discussing it.

**Quality:**

3

**Strengths And Weaknesses:**

Strengths

The method is simple and easy to implement, even without sample code. It is flexible and can be applied to arbitrary groupings, such as filters, neurons, or attention heads. This flexibility may allow it to be adapted to specific hardware constraints or architectural needs.

Although the theoretical results are developed under SGD, the method appears to work with other optimizers in practice, as shown in the NanoGPT experiments using Adam.

Figures 1 and 2 show a clear attempt to build reader intuition around the method and the gating structure.

Weaknesses

The paper does not clearly quantify the overhead of the method. While it states that the parameter overhead is negligible, this should be more precise. For example, how many additional parameters are introduced in each experiment (calculable as D x J)? More importantly, what is the compute overhead? Each primary parameter appears to require D extra multiplications during forward and backward passes. How does this compare to the total FLOPs per token in standard models like Transformers? The authors could have provided FLOP estimates or at least wall-clock comparisons with and without D-Gating.

There is a mismatch between the theoretical and practical setups. The theory is developed for SGD, but the LLM experiments use Adam, which has different optimization dynamics. If the key convergence properties rely on SGD-specific behavior, the authors should either justify the use of Adam or empirically validate that the same convergence trends (or group balance statistics) hold under Adam.

The exposition is weaker than it should be. The key idea is simple, but the paper takes too long to explain it clearly. Even the abstract uses terms like "overparameterization" and "structured sparsity" without clarifying that the focus is on group sparsity applied to network components like heads or filters.

Figure 2 may not be the clearest way to introduce the method. It shows input-wise grouping, which could mislead readers about the method’s generality (or, for me on a first read, how it actually operates). A different grouping example might have been more effective, or at least a caption that explains the meaning of the colors.

The paper loosely equates loss convergence with "balancedness" in ways that are not always intuitive or validated empirically. For example, Equation 7 defines balancedness as equal norms within each group, but what about balance across groups? Figure 3 claims to show imbalance decay, but it only plots loss convergence, not any norm-based imbalance metric. Since the paper equates balancedness with loss convergence theoretically, it would be helpful to empirically track both rather than rely on the theoretical connection.

The experiments are on a very small scale. For instance, the NanoGPT experiments use the Shakespeare dataset, which is tiny relative to modern LLMs. This suggests that D-Gating may introduce overheads that would make it impractical at larger scales (see points above). The paper also lacks strong baselines. For example, what happens if we just train a smaller dense model with fewer parameters? Would that reach the same loss with fewer total FLOPs? The benefit of D-Gating is supposed to be dynamic sparsity during training, but this should be framed in terms of loss vs. total FLOPs, as that is the key metric used in the LLM community.  Putting aside whether we can actually achieve high utilization with the FLOPs varying dynamically throughout training is already a simplication.

Finally, the related work section could be strengthened by including a table that highlights how D-Gating compares to prior methods. Columns like "compatible with SGD", "architecture-agnostic", and "fully differentiable" would help readers see the method’s advantages more clearly.

---

> ### Author Rebuttal · Authors · 2025-07-29
>
> Dear Reviewer 4J3D,
>
> Thank you for your time and effort to evaluate our submission, as well as your interesting comments and questions. In the following, we first respond to the weaknesses and then address the limitations and questions.
>
> # Weaknesses
>
> ### **(W1)**: no overhead quantification
>
> We agree that these experiments would corroborate the utility of our method in practical settings. To this end, we included several additional experiments on **a)** the wall-clock runtime during training for different models, batch sizes, and $D$, **b)** the GPU memory overhead for the same settings, and **c)** computed the parameter overhead in our models as you suggested. Generally, the **parameter and memory overhead is negligible** for any batch size, while the **wall-clock runtime increases slightly with $D$ for some models**, but this difference becomes irrelevant for larger batch sizes.
>
> ------
>
> **Parameter Overhead (factor=new params/old params -1)**
>
> | Model | Params| Add. Params | Factor ($D=4$)| Structure|
> |--------|---------|------------|--------|------|
> | Lin. Mod. |200|40⋅(D-1) | $6.0 \times 10^{-1}$  | 40 groups of 5 features    |
> | LeNet-300-100| ≈266k |input dim⋅(D-1)  | $8.8 \times 10^{-3}$  | Input dim $\in [400, 784]$  |
> | VGG-16| ≈15mio. |4,224⋅(D-1) | $8.4 \times 10^{-4}$   | Sum of conv. filters |
> | NanoGPT| ≈10.8mio.|72⋅(D-1)| $2 \times 10^{-5}$ | 6 layers, 12 heads |
>
> ------
>
> **Peak memory utilization during training (GB). Means and std. over ten runs are shown.**
>
> |mem. (GB) |D|BS=128 |BS=256 |BS=512 |BS=1024 |
> |---------------|--|----------------|----------------|----------------|----------------|
> |**VGG-16** |1|1.2449 (0.5116)|1.4715 (0.2340)|1.4967 (0.1974)|1.6416 (0.8027)|
> | |2|1.0844 (0.1329)|1.6644 (0.3408)|1.3656 (0.1662)|1.5073 (0.2974)|
> | |3|0.9857 (0.1211)|1.4344 (0.1882)|1.4172 (0.1496)|1.5120 (0.2128)|
> |  |4|0.9467 (0.1687)|1.3983 (0.1875)|1.4512 (0.2493)|1.3377 (0.1059)|
>
> |mem. (GB) |D|BS=16 |BS=32 |BS=64 |
> |---------|--|------------------|------------------|------------------|
> |**NanoGPT**|1|1.4060 (0.0232)|2.6474 (0.0417)|5.1592 (0.0460)|
> | |2|1.4060 (0.0216)|2.6570 (0.0360)|5.1370 (0.0571)|
> | |3|1.4272 (0.0192)|2.6571 (0.0348)|5.1096 (0.0559)|
> | |4|1.4152 (0.0220)|2.6627 (0.0456)|5.0993 (0.0698)|
>
> ------
>
> **Wall-clock runtime overhead (training) for neuron, filter, and attention sparsity. Means and std. over ten runs are shown.**
>
> | ms/sample| D | BS=128  | BS=256 | BS=512| BS=1024 |
> |----------------|---|----------------|----------------|----------------|----------------|
> |**LeNet-300-100**| 1| 0.0131 (0.0003)| 0.0072 (0.0002)| 0.0046 (0.0003)| 0.0030 (0.0003)|
> | | 2 | 0.0151 (0.0003)| 0.0082 (0.0002)| 0.0051 (0.0004)| 0.0032 (0.0000)|
> | | 3 | 0.0158 (0.0002)| 0.0093 (0.0008)| 0.0050 (0.0001)| 0.0046 (0.0011)|
> |  | 4 | 0.0170 (0.0015)| 0.0090 (0.0002)| 0.0052 (0.0001)| 0.0041 (0.0009)|
> |**VGG-16** | 1 | 0.3111 (0.0089)| 0.3054 (0.0126)| 0.1587 (0.0062)| 0.0796 (0.0038)|
> | | 2 | 0.3213 (0.0071)| 0.3066 (0.0066)| 0.1576 (0.0039)| 0.0835 (0.0027)|
> | | 3 | 0.3098 (0.0073)| 0.3019 (0.0013)| 0.1542 (0.0035)| 0.0804 (0.0016)|
> | | 4 | 0.3159 (0.0167)| 0.3057 (0.0084)| 0.1573 (0.0041)| 0.0796 (0.0021)|
>
> | ms/sample | D | BS = 16 | BS = 32 | BS = 64 |
> |--------|---|----------------|--------|-------------|
> |**NanoGPT**| 1 | 0.0508 (0.0012)| 0.0266 (0.0006)| 0.0174 (0.0004)|
> | | 2 | 0.0603 (0.0011)| 0.0314 (0.0006)| 0.0185 (0.0002)|
> || 3 | 0.0606 (0.0009)| 0.0314 (0.0007)| 0.0186 (0.0002)|
> | | 4 | 0.0617 (0.0017)| 0.0317 (0.0009)| 0.0187 (0.0002)|
>
> ------
>
> ### **(W2)**: mismatch between theoretical and practical set-up
>
> Thanks for pointing out this subtle point. Our main result, Theorem 1, establishes that our $D$-Gating objective is equivalent to the non-smooth sparsity-regularized base objective, irrespective of the optimizer. This equivalence in terms of the solution structure permits, in principle, using any optimizer to find those sparse solutions. Our convergence result, Lemma 4, quantifying how fast the $D$-Gated objective aligns with the non-smooth sparse objective, merely provides a clean upper bound for the rate under gradient flow. Hence, there is no mismatch between our theoretical and practical set-up, just that the convergence rate under gradient flow does not hold for, e.g., Adam.
> Intuitively, any common optimizer strives to find a balanced representation, since, at any point, the $D$-Gating parameters can be perturbed toward a more balanced solution, thus reducing the $L_2$ penalty, while leaving the unregularized loss constant. To corroborate this intuition, we repeated the experiment in Fig.3 using Adam, not SGD. We find that the imbalance also decays to zero (even faster than SGD), although the decay is not exponential. This indicates that although the rate may not be as cleanly described as for gradient flow, Adam also leads to balanced gating representations and thus a sparsity-inducing objective.
>
> ### **(W3)**: use of words in exposition
>
> We agree that “structured sparsity” can also encompass, e.g., N:M sparsity, not covered by our group sparse framework, although it is understood as group sparsity in certain communities. The phrase overparameterization was used as our $D$-Gating approach is inherently a (regularized) overparameterization. We will diligently go through our submission and reword it to emphasize that our approach is for group sparsity in a revised version.
>
> ### **(W4)**: visualization of our method (Fig.2)
>
> Thank you for pointing this out. We also spent quite some time deliberating the best way to visualize our approach in a simple way and decided on the presented input-sparse example with explicitly mentioning that this is only one possible use case in the caption. Note that our submission also includes more visualizations of $D$-Gating architectures in Figures 9, 10, and 13 of our Appendix. Yet, we will reconsider our example choice and try to come up with a better visualization for a revised version.
>
> ### **(W5)**: Connection between loss convergence, norm regularizers, and balancedness
>
> For clarity, we have added a version of Fig. 3 to the Appendix that shows the difference between the $L_{2}$ and the $L_{2,2/D}$ regularizer on the y-axis, showing qualitatively identical results. This is because the loss difference is simply the difference of both norm-based regularizers scaled by $\lambda$. To see this, $\mathcal{L}(\boldsymbol{\omega}, \boldsymbol{\Gamma}) - \mathcal{L}\_{\mathbf{w}}(\mathbf{w}) = \mathcal{L}\_0(\cdot) -  \mathcal{L}\_0(\cdot) + \lambda \big( (||\boldsymbol{\omega}|_2^2| + || \boldsymbol{\Gamma}||_F^2)/D - ||\mathbf{w}||\_{2,2/D}^{2/D} \big)$. Hence, convergence of losses is virtually identical to convergence of the regularizers, which happens iff all parameters are balanced.
>
> ### **(W6)**: The experiments are on a very small scale.[...] This suggests that D-Gating may introduce overheads that would make it impractical at larger scales. The paper also lacks strong baselines.
>
> Regarding the overheads, please see our new experiments described in **(W1)**, showing that they are small/negligible.
>
> Regarding the scale of experiments, while we agree that large-scale LLM experiments would strengthen our evaluation, we argue that our contribution has a methodological focus, and the experiments serve to showcase versatility. Given the decision space {architecture types}x{data modalities}x{sparsity use cases} is quite large, we tried to best represent this in our selection of experiments, spanning fully-connected DNNs, CNNs, neural trees, and transformers, using numerical, image, text, and speech data of various sizes, and successfully demonstrates $D$-Gating for input sparsity, filter sparsity, tree sparsity, and attention sparsity.
>
> Regarding baselines, we agree that they are solid but dated methods, particularly for transformers where alternatives like SparseGPT exist. However, our contribution is mainly methodological, proposing a differentiable formulation of $L_{2,2/D}$ regularization, with the experiments demonstrating its effectiveness for various architectures and data modalities. SparseGPT, for example, is a method that efficiently solves a weight reconstruction problem post-training under sparsity constraints, and is thus fundamentally different from our method. Hence, we selected more directly comparable baselines for our experiments, i.e., direct optimization of the non-smooth $L_{2,1}$ regularizer (besides Network Slimming and LassoNet/HSIC for the feature selection experiments), to better illustrate our method’s advantage over the naive formulation, instead of trying to establish SOTA performance for every application.
> Moreover, we note that most SOTA methods are multi-pass, pruning-and-regrowing, or post-training weight reconstruction strategies that make a fair comparison much harder (see [1]).
>
> # Limitations/Questions
>
> ### **(Q1)**: key distinction between "modular" and "versatile" ?
>
> We agree it can sound redundant and changed the title to “Differentiable Sparsity via $D$-Gating: Simple and Versatile Structured Penalization”.
>
> ### **(L1)**: [...] theoretical assumptions that may be violated in practice, e.g., in the very experiments used in the paper!
>
> We hope the previous discussion has cleared up this misunderstanding. The equivalence of the $D$-Gating objective holds for any optimizer, just the exponential rate we derived is for gradient flow specifically.
>
> ### **(L2)**: Discussion of computational overheads was inadequate, and it should be clearly quantified, rather than simply saying "negligible"
>
> Please see response to **(W1)**.
>
> ------
>
> We thank you again and hope this adequately answers your questions and concerns. We also hope that our additional experimental and theoretical results strengthen our submission in your eyes.
>
> References:
>
> [1] Wang, Huan, et al. "Why is the state of neural network pruning so confusing? on the fairness, comparison setup, and trainability in network pruning." arXiv:2301.05219 (2023).

---

> > ### Comment · Reviewer_4J3D · 2025-08-04
> > **Response to rebuttal**
> >
> > I have read the other reviews and rebuttals, and the rebuttal to my own review.  I feel the responses help improve the clarity and quality of the work, and will adjust my score.

---

> > > ### Author Response · Authors · 2025-08-08
> > >
> > > Dear Reviewer 4J3D,
> > >
> > > Thank you again for your time and detailed feedback, as well as for considering our rebuttals to the other reviews. We appreciate that the additional experiments and explanations in our responses improved the clarity and quality of our submission in your eyes, and thank you for adjusting your score accordingly.

---

### Official Review · Reviewer_vx9E · 2025-06-25

**Clarity:** 2
**Significance:** 2
**Originality:** 3
**Rating:** 4
**Confidence:** 3

**Summary:**

This paper proposes a new way of imposing sparsity regularization to neural networks named as D-gating, achieved from combining a model re-parameterization trick with $L_2$ regularization. The authors argue that D-gating is modular and compatible with arbitrary architectures and show its equivalence with a generalized version of group Lasso regularization. Theoretically, the convergence speed along D-gating optimization trajectory is analyzed. Empirically, the authors conduct experiments in various downstream tasks  (feature selection, filter sparsity, language modeling and neural tree sparsity) an demonstrate the modularity and performance superiority of D-gating.

**Questions:**

1. I am not an expert in this field, but the experimental results seem to only compare D-gating with a few standard sparsity regularization methods such as Group Lasso or network-slimming. Can you include more in-depth comparisons with newer (structured and unstructured) methods such as [1,2]?
2. Can you provide theoretical guarantees for convergence of SGD to sparse solutions (See the strengths and weaknesses section)?

References:
[1]. Lasby, Mike, et al. "Dynamic sparse training with structured sparsity." arXiv preprint arXiv:2305.02299 (2023).
[2]. Tai, Kai Sheng, Taipeng Tian, and Ser Nam Lim. "Spartan: Differentiable sparsity via regularized transportation." Advances in Neural Information Processing Systems 35 (2022): 4189-4202.

**Ethical Concerns:**

["NO or VERY MINOR ethics concerns only"]

**Final Justification:**

I have read the author rebuttal and other reviews. Since thorough experiments are provided to address my concern on effectiveness of the method, I have raised my score accordingly.

**Limitations:**

Yes.

**Quality:**

3

**Strengths And Weaknesses:**

Quality:

I am not concerned about the soundness of theoretical proofs given their standard derivations. The empirical results also seem fine to me.

Clarity:

The general writing is fine. However, the technical notation system sometimes introduces unnecessary complications. For example, the D-gating framework, which is a direct re-parameterization introducing $D-1$ additional multiplicative parameters, is expressed in a complex notation system without much explanation. I suggest the authors simplify their technical writing and add more explanations to improve readability of the paper.

Significance:

The idea of achieving differentiable sparsity regularization via D-gating reparameterization seems interesting to me. The theoretical proofs are standard and derived from routine optimization analysis, but the authors also demonstrate the performance of their method with empirical validation.

Nevertheless, I think this paper has the following weaknesses:
1. While the authors claim that one major advantage of D-gating is its modularity and compatibility with SGD, the theoretical analysis for SGD (lemma 4) relies on a strong assumption: a balanced zero representation can be achieved. The more important part
question in optimization trajectory: how SGD reaches such a zero state, is unguaranteed.
2. The authors claim the parameter overhead from the gating factors is negligible. However, the additional computational cost can be large with large $D$ or models with a very large number of small groups.

Originality:

To the best of my knowledge, the D-gating framework introduced in the paper is new and original,

---

> ### Author Rebuttal · Authors · 2025-07-29
>
> Dear Reviewer vx9E,
>
> Thank you for your time and effort to evaluate our submission, as well as your interesting questions. In the following, we first respond to the weaknesses and then address the questions.
>
> # Weaknesses
>
> ### **(W1)**: how SGD reaches such a zero (imbalance) state, is unguaranteed
>
> Please see our response to question (Q2) below. Further, note that all our experiments also support the claim that SGD finds sparse solutions. E.g., in Fig. 4, we show that $D$-Gating finds exactly the same global minima as the Group Lasso for linear models.
>
> ### **(W2)**: Additional computational cost can be large
>
> We agree with you that the parameter count overhead could become considerable for extremely few weights, small groups, and many/all parameters being regularized. But in practice, the **param overhead is negligible**. Even for the simple linear regression with 200 features and 40 groups, $D$-Gating with $D=4$ adds only 120 new parameters to the model. In contrast, for our NanoGPT implementation, we use 6 layers with 12 heads, yielding 72 groups in total. For $D=4$ **this amounts to 216 additional parameters in a model with 10.8 mio. parameters**. To illustrate this better, we have included the following table in the Appendix:
>
> ------
>
> **Paramever Overhead (factor=new params/old params-1)**
>
> | Model | Params| Add. Params | Factor ($D=4$)| Details|
> |-----|--------|------|------|-----|
> | Lin. Mod. | 200 |40⋅(D-1) | $6.0 \times 10^{-1}$  | 40 groups of 5 features    |
> | LeNet-300-100 | ≈266k | input dim⋅(D-1)  | $8.8 \times 10^{-3}$  | Input dim $\in [400, 784]$  |
> | VGG-16 | ≈15mio. | 4,224⋅(D-1) | $8.4 \times 10^{-4}$   | Sum of conv. filters |
> | NanoGPT | ≈10.8mio.|72⋅(D-1)| $2 \times 10^{-5}$ | 6 layers, 12 heads |
>
> ------
>
> In addition, to verify that the computational overhead is not considerable, we measured the **wall-clock runtime** and **GPU memory overhead** for different models and batch sizes, summarized in the tables below. **Memory overhead is negligible** and **runtime slightly increases in $D$** but only for small batches.
>
> **Peak memory utilization during training (GB). Means and std. over ten runs are shown.**
>
> |mem. (GB) |D|BS=128 |BS=256 |BS=512 |BS=1024 |
> |------|--|------|------|------|--------|
> |**VGG-16**   |1|1.2449 (0.5116)|1.4715 (0.2340)|1.4967 (0.1974)|1.6416 (0.8027)|
> | |2|1.0844 (0.1329)|1.6644 (0.3408)|1.3656 (0.1662)|1.5073 (0.2974)|
> | |3|0.9857 (0.1211)|1.4344 (0.1882)|1.4172 (0.1496)|1.5120 (0.2128)|
> |  |4|0.9467 (0.1687)|1.3983 (0.1875)|1.4512 (0.2493)|1.3377 (0.1059)|
>
> |mem. (GB) |D|BS=16 |BS=32 |BS=64 |
> |-------|--|-------|------|------|
> |**NanoGPT**|1|1.4060 (0.0232)|2.6474 (0.0417)|5.1592 (0.0460)|
> | |2|1.4060 (0.0216)|2.6570 (0.0360)|5.1370 (0.0571)|
> | |3|1.4272 (0.0192)|2.6571 (0.0348)|5.1096 (0.0559)|
> | |4|1.4152 (0.0220)|2.6627 (0.0456)|5.0993 (0.0698)|
>
> ------
>
> **Wallclock runtime overhead (training) for neuron, filter, and attention sparsity. Means and std. over ten runs are shown.**
>
> | ms/sample| D | BS=128  | BS=256 | BS=512| BS=1024 |
> |-----|---|------|-----|-----|--------|
> |**LeNet-300-100**| 1 | 0.0131 (0.0003)| 0.0072 (0.0002)| 0.0046 (0.0003)| 0.0030 (0.0003)|
> | | 2 | 0.0151 (0.0003)| 0.0082 (0.0002)| 0.0051 (0.0004)| 0.0032 (0.0000)|
> | | 3 | 0.0158 (0.0002)| 0.0093 (0.0008)| 0.0050 (0.0001)| 0.0046 (0.0011)|
> |   | 4 | 0.0170 (0.0015)| 0.0090 (0.0002)| 0.0052 (0.0001)| 0.0041 (0.0009)|
> |**VGG-16** | 1 | 0.3111 (0.0089)| 0.3054 (0.0126)| 0.1587 (0.0062)| 0.0796 (0.0038)|
> | | 2 | 0.3213 (0.0071)| 0.3066 (0.0066)| 0.1576 (0.0039)| 0.0835 (0.0027)|
> | | 3 | 0.3098 (0.0073)| 0.3019 (0.0013)| 0.1542 (0.0035)| 0.0804 (0.0016)|
> | | 4 | 0.3159 (0.0167)| 0.3057 (0.0084)| 0.1573 (0.0041)| 0.0796 (0.0021)|
>
> | ms/sample | D | BS = 16 | BS = 32 | BS = 64 |
> |-----|---|------|----|----|
> |**NanoGPT**| 1 | 0.0508 (0.0012)| 0.0266 (0.0006)| 0.0174 (0.0004)|
> |  | 2 | 0.0603 (0.0011)| 0.0314 (0.0006)| 0.0185 (0.0002)|
> || 3 | 0.0606 (0.0009)| 0.0314 (0.0007)| 0.0186 (0.0002)|
> | | 4 | 0.0617 (0.0017)| 0.0317 (0.0009)| 0.0187 (0.0002)|
>
> ------
>
> # Questions
>
> ### **(Q1)**: no in-depth comparisons with SOTA methods
>
> While we agree that a comparison with sophisticated sparsification pipelines would be interesting, our contribution is mainly methodological, proposing a differentiable formulation of $L_{2,2/D}$ regularization, with the experimental evaluation demonstrating its effectiveness for various architectures and data modalities. For example, the method SRigL proposed in your reference [1] is a specialized approach for N:M-type structured sparsity with a constant fan-in constraint and based on pruning and re-growing connections, which is fundamentally different from our differentiable sparsity regularization approach. Therefore, we chose to opt for more directly comparable baselines in our experiments, i.e., direct optimization of the non-smooth $L_{2,1}$ regularizer, Network Slimming, or LassoNet, to better illustrate our method’s advantage over the naive formulation, instead of trying to establish SOTA performance for every application. Moreover, as shown in [2], the comparison set-up is massively complicated for sophisticated pruning and retraining/regrowing pipelines, further corroborating our use of directly related methods, albeit they may not be SOTA.
>
> ### **(Q2)**: Can you provide theoretical guarantees for convergence of SGD to sparse solutions?
>
> We thank you for this interesting question, which highlights a subtle difference between convergence of the objective to a solution and convergence of the objective $\mathcal{L}(\boldsymbol{\omega}, \boldsymbol{\Gamma})$ to the non-smooth objective ${\mathcal{L}}_{\mathrm{w}}(\mathrm{w})$. Whether a $D$-Gated network converges to a solution overall (i.e., to solutions with many zero groups) can not be answered simply, as it depends on many factors like the network architecture, data set, and optimization hyperparameters. It is similar to asking whether objectives with explicit Group Lasso regularization always converge, which can not be answered without specifying the concrete learning problem that is being regularized. A formal analysis of how $D$-Gating impacts overall convergence rates is beyond the scope of this paper, evidenced by the lengthy but still restricted derivations in [3], obtaining the Kurdyka-Łojasiewicz exponent for a much simpler (unstructured) overparameterization, from which a local convergence rate can be deducted.
> In contrast, our theoretical results establish that a) all solutions of the $D$-Gating problem are solutions to the sparse regularized problem, and b) under gradient flow, the surrogate objective exponentially fast becomes the sparse regularized objective, i.e., the optimization of $\mathcal{L}(\boldsymbol{\omega}, \boldsymbol{\Gamma})$ turns into $\mathcal{L}\_{\mathbf{w}}(\mathbf{w})$.
>
> ### **Other comments**:
>
> #### **New result for discrete-time SGD dynamics**
> Additionally, we have now extended our gradient flow results to discrete-time gradient descent, establishing (for sufficiently small step sizes) exponential decay of the imbalance similar to gradient flow, and thus $\mathcal{L}(\boldsymbol{\omega}, \boldsymbol{\Gamma}) \to {\mathcal{L}}_{\mathrm{w}}(\mathrm{w})$. The novel Lemma reads:
>
> > Consider the $D$-Gated objective Eq. (5). Then, under (S)GD with learning rate $\eta>0$, (i) the pair-wise imbalances in Eq. (13) evolve as $\mathcal{I}\_{j,d,d'}^{(t+1)} = \big(1 - \frac{4\lambda}{D} \eta\big) \mathcal{I}\_{j,d,d'}^{(t)} + \eta^2 \Delta\_{j,d,d'}^{(t)}, \\: d \neq d', d,d' \in [D]$, with different second-order terms $\Delta\_{j,d,d'}$ for $\mathcal{I}\_{j,1,d'}$ and $\mathcal{I}\_{j,d,d'}$ for $d,d'>1$. In particular, for sufficiently small $\eta$, $\mathcal{I}_{j,d,d'}^{(t+1)} \approx (1-4 \lambda \eta/D) \cdot  \mathcal{I}\_{j,d,d'}^{(t)}$ exhibits discrete exponential decay. Moreover, (ii) for balanced zero gating parameters, $(\boldsymbol{\omega}_j^{(t)},\{\gamma\_{j,d}^{(t)}\}\_{d=1}^{D-1})=\boldsymbol{0}$, the future iterates are confined to $\mathbf{w}_j^{(t')}=\boldsymbol{0} \, \forall \,t'>t$, and (iii), balancedness, i.e.,  $\mathcal{I}\_{j,d,d'}^{(t)}=0$, is conserved $\forall t'>t$ between any two scalar gates $\gamma\_{j,d}^{(t)},\gamma\_{j,d'}^{(t)}$.
>
> #### **Choice of notation**
> We agree with you that it is not straightforward to read, but after much deliberation, we chose this notation to emphasize the “gating” operation using the triangle. The key issue is to find a representation for “group-wise element-wise multiplication”, a problem that other authors have also faced. E.g., [4] introduces an additional matrix $D$ that contains $0$s and $1$s to “broadcast” the group factors to the same dimension as the primary weight, yielding the expression $(D \boldsymbol{\gamma}^{\odot} ) \odot  \boldsymbol{\omega}$, which is arguably even more inconvenient, while [5] use the symbol $\boldsymbol{\omega }\odot_{\mathcal{G}} (\boldsymbol{\gamma}^{\odot})$ for structured element-wise multiplication. But agreeing with your sentiment, we will think about how to best simplify the notation for a revised version, nevertheless.
>
> ------
>
> We thank you again and hope this adequately answers your questions. We also hope that the additional experimental results, as well as our new theoretical result for the discrete-time evolution of imbalances, strengthen our submission in your eyes.
>
> References
>
> [1] Lasby, M., et al. "Dynamic Sparse Training with Structured Sparsity." The 12th ICLR.
>
> [2] Wang, H., et al. "Why is the state of neural network pruning so confusing? on the fairness, comparison setup, and trainability in network pruning." arXiv:2301.05219 (2023).
>
> [3] Ouyang, W., et al. "Kurdyka–Łojasiewicz exponent via Hadamard parametrization." SIAM Journal on Optimization 35.1 (2025): 62-91.
>
> [4] Li, Jiangyuan, et al. "Implicit Regularization for Group Sparsity." 11th ICLR.
>
> [5] Poon, C., and Gabriel P. "Smooth bilevel programming for sparse regularization." NeurIPS 34 (2021)

---

> > ### Comment · Reviewer_vx9E · 2025-08-04
> >
> > Thank you for the thorough experiments demonstrating D-gating's efficiency. I understand that a rigorous theoretical analysis and comparison with domain-specific SOTA might extend beyond the scope of your current research, although such results would strengthen the work considerably. I will raise my score to 4.

---

> > > ### Author Response · Authors · 2025-08-08
> > >
> > > Dear Reviewer vx9E,
> > >
> > > We thank you again for your time and constructive feedback, which valuably helped to improve our submission in important aspects by adding experiments demonstrating the efficiency and low overhead of $D$-Gating, as well as for subsequently raising your score.

---

### Official Review · Reviewer_dzXs · 2025-07-02

**Clarity:** 4
**Significance:** 4
**Originality:** 3
**Rating:** 6
**Confidence:** 4

**Summary:**

The authors propose an approach to induce structured sparsity during training through D gating without requiring additional passes for pruning. The approach is compatible with SGD optimization and shows that the method is appliable on different taks including attention based language model, where sparsification is more challenging.

**Questions:**

So far i understood the method yields to overparametrization. Deeper gating based on the figures 5,6 delivers the best sparsity–accuracy trade-off, yet adds parameters and potentially slows convergence. Can you quantify the memory/computation overhead of increasing *D* and outline a guideline for choosing the gate depth in practice?

**Ethical Concerns:**

["NO or VERY MINOR ethics concerns only"]

**Final Justification:**

Additional experiments showed that D gating add minimal to no overhead, even for the recommended values of 3 and 4

**Limitations:**

Yes

**Quality:**

4

**Strengths And Weaknesses:**

### Strengths
1. **Zero-overhead, SGD-compatible formulation**
   D-Gating is implemented as a smooth regulariser; it requires no custom solvers, proximal steps, or pruning–retrain cycle, so it drops into vanilla SGD/AdamW pipelines with virtually no engineering overhead.

2. **Task-agnostic applicability**
   The paper demonstrates the same gating mechanism on image classification (VGG-16, ResNet-18) *and* language modelling (NanoGPT), suggesting that the approach generalises across modalities.

3. **Superior extreme-sparsity performance**
   With deeper gating ( *D = 4* ), the method outperforms L₂,₁ + magnitude pruning and plain magnitude pruning at the highest sparsity regimes for both CNNs and NanoGPT heads, where baselines usually collapse.

4. **Solid theoretical grounding**
   The authors supply clear, formal proofs that the differentiable surrogate shares its local minima with the non-smooth L₂,₂∕ᴰ objective and that unbalanced gate factors vanish exponentially fast.

5. **Meaningful proxy for real acceleration**
   Although no wall-clock inference is reported, FLOP estimates are provided to quantify *potential* speed-ups, giving readers a first-order sense of deployability.

4. The paper is very well written and the appendix is structured to support the claims made.
---

### Weaknesses
1. **Restricted model and dataset scope**
   Vision experiments stop at VGG-16/ResNet-18 on relatively small benchmarks; NLP results are limited to NanoGPT. Evaluating on a transformer vision model (e.g., ViT-S) or larger datasets would better establish robustness.

2. **Baselines are not state of the art**
   Magnitude pruning and Network Slimming are solid but dated. More recent structured-sparsity methods (e.g., GMP, SparseGPT, DynaSparse) could provide a tougher comparative bar.

3. **Theoretical speed-up unverified in practice**
   All acceleration claims are FLOP-based; without end-to-end latency or energy numbers on real hardware, the practical benefit remains speculative.

4. **No ablation on gate depth vs. overhead**
   Deeper gating (*D ≥ 3*) yields higher sparsity, but the paper does not quantify the additional parameter/memory cost or convergence stability trade-off.

---

> ### Author Rebuttal · Authors · 2025-07-29
>
> Dear Reviewer dzXs,
>
> Thank you for your encouraging comments and the time and effort to evaluate our submission. In the following, we first respond to the question and then address the weaknesses.
>
> # Questions
>
> ### **(Q1)**: Can you quantify the memory/computation overhead of increasing D?
>
> We agree that those experiments would help in evaluating practical implementation considerations. To this end, we have included several additional experiments on **a)** the wall-clock runtime overhead during training for different models, batch sizes, and $D$, **b)** the GPU memory overhead for the same settings, and **c)** computed the parameter overhead in our models. Generally, the **parameter and memory overhead are essentially negligible** for any batch size, while the **wall-clock runtime increases slightly with $D$ for some models**. However, this difference becomes irrelevant for larger batch sizes.
>
> ------
>
> **Wallclock runtime overhead (training) for neuron, filter, and attention sparsity. Means and std. over ten runs are shown.**
>
> | ms/sample| D | BS=128  | BS=256 | BS=512| BS=1024 |
> |----------------|---|----------------|----------------|----------------|----------------|
> |**LeNet-300-100**| 1 | 0.0131 (0.0003)| 0.0072 (0.0002)| 0.0046 (0.0003)| 0.0030 (0.0003)|
> | | 2 | 0.0151 (0.0003)| 0.0082 (0.0002)| 0.0051 (0.0004)| 0.0032 (0.0000)|
> | | 3 | 0.0158 (0.0002)| 0.0093 (0.0008)| 0.0050 (0.0001)| 0.0046 (0.0011)|
> |   | 4 | 0.0170 (0.0015)| 0.0090 (0.0002)| 0.0052 (0.0001)| 0.0041 (0.0009)|
> |**VGG-16**      | 1 | 0.3111 (0.0089)| 0.3054 (0.0126)| 0.1587 (0.0062)| 0.0796 (0.0038)|
> | | 2 | 0.3213 (0.0071)| 0.3066 (0.0066)| 0.1576 (0.0039)| 0.0835 (0.0027)|
> | | 3 | 0.3098 (0.0073)| 0.3019 (0.0013)| 0.1542 (0.0035)| 0.0804 (0.0016)|
> | | 4 | 0.3159 (0.0167)| 0.3057 (0.0084)| 0.1573 (0.0041)| 0.0796 (0.0021)|
>
> | ms/sample | D | BS = 16 | BS = 32 | BS = 64 |
> |-----------|---|----------------|----------------|----------------|
> |**NanoGPT**| 1 | 0.0508 (0.0012)| 0.0266 (0.0006)| 0.0174 (0.0004)|
> |  | 2 | 0.0603 (0.0011)| 0.0314 (0.0006)| 0.0185 (0.0002)|
> || 3 | 0.0606 (0.0009)| 0.0314 (0.0007)| 0.0186 (0.0002)|
> | | 4 | 0.0617 (0.0017)| 0.0317 (0.0009)| 0.0187 (0.0002)|
>
> ------
>
> **Peak memory utilization during training (GB). Means and std. over ten runs are shown.**
>
> |mem. (GB) |D|BS=128 |BS=256 |BS=512 |BS=1024 |
> |---------------|--|----------------|----------------|----------------|----------------|
> |**VGG-16**   |1|1.2449 (0.5116)|1.4715 (0.2340)|1.4967 (0.1974)|1.6416 (0.8027)|
> | |2|1.0844 (0.1329)|1.6644 (0.3408)|1.3656 (0.1662)|1.5073 (0.2974)|
> | |3|0.9857 (0.1211)|1.4344 (0.1882)|1.4172 (0.1496)|1.5120 (0.2128)|
> |  |4|0.9467 (0.1687)|1.3983 (0.1875)|1.4512 (0.2493)|1.3377 (0.1059)|
>
> |mem. (GB) |D|BS=16 |BS=32 |BS=64 |
> |---------|--|---------|----------|--------|
> |**NanoGPT**|1|1.4060 (0.0232)|2.6474 (0.0417)|5.1592 (0.0460)|
> | |2|1.4060 (0.0216)|2.6570 (0.0360)|5.1370 (0.0571)|
> | |3|1.4272 (0.0192)|2.6571 (0.0348)|5.1096 (0.0559)|
> | |4|1.4152 (0.0220)|2.6627 (0.0456)|5.0993 (0.0698)|
>
> ------
>
> **Parameter Overhead (factor=new params/old params-1)**
>
> | Model | Params| Add. Params | Factor ($D=4$)| Details|
> |------------------|---------|-------------|------------|---------------------|
> | Lin. Mod. | 200 |40⋅(D-1) | $6.0 \times 10^{-1}$  | 40 groups of 5 features    |
> | LeNet-300-100 | ≈266k | input dim⋅(D-1)  | $8.8 \times 10^{-3}$  | Input dim $\in [400, 784]$  |
> | VGG-16 | ≈15mio. | 4,224⋅(D-1) | $8.4 \times 10^{-4}$   | Sum of conv. filters |
> | NanoGPT | ≈10.8mio.|72⋅(D-1)| $2 \times 10^{-5}$ | 6 layers, 12 heads |
>
> ------
>
> ### **(Q2)**: Can you outline a guideline for choosing the gate depth in practice?
>
> In the general non-convex regularization literature [1], it is established that $L_{2,0.5}$ or $L_{2,2/3}$ regularization ($D=3,4$) usually yields the best trade-off between performance and increased optimization issues as we approach group $L_0$ regularization.
>
> We conducted ablation studies on $D$ for group sparse linear regression, an input-sparse LeNet-300-100 trained on ISOLET, and a filter-sparse VGG-16 trained on CIFAR-10. As expected, some settings showed increased **numerical instability for large $D$** (a jagged/“zig-zag” regularization path instead of a smooth one) and all settings confirmed that **$D=4$ essentially reaps the full benefits of non-convex regularization** while retaining stable training. The new plots are summarized as tables below.
>
> ------
>
> **Ablation studies for gating depth $D$ at various sparsity levels s (either test acc. or RMSE)**
>
> | LeNet (ISOLET) | s=40% | s=60% | s=80% | s=90% | s=92% | s=94% | s=96% | s=98% |
> |----------|-------|-------|-------|-------|-------|-------|-------|-------|
> | 2-Gating | 95.00%| 93.66%| 68.53%| 49.81%| 46.07%| 42.32%| 38.23%| 21.04%|
> | 3-Gating | 94.89%|**94.97%**|**94.81%**|**94.81%**|93.95%|92.56%|87.28%|70.57%|
> | 4-Gating |**95.09%**|94.59%|94.27%|93.30%|**93.40%**|**92.75%**|**89.57%**|74.57%|
> | 8-Gating | 94.75%|94.09%|93.42%|92.81%|92.49%|92.32%|87.90%|**76.39%**|
> |10-Gating | 94.54%|93.76%|92.99%|92.76%|92.64%|92.24%|88.76%|73.74%|
> |20-Gating | 94.24%|93.51%|92.78%|92.53%|92.11%|92.19%|86.28%|61.31%|
>
> | VGG16 | s=40% | s=60% | s=80% | s=90% | s=92% | s=94% | s=96% | s=98% |
> |----------|-------|-------|-------|-------|-------|-------|-------|-------|
> | 2-Gating | 93.11%| 92.96%| 90.74%| 88.24%| 86.54%| 84.20%| 78.62%| 47.54%|
> | 3-Gating |**93.58%**|**93.39%**|91.55%|88.14%|87.10%|84.57%|82.04%|73.55%|
> | 4-Gating |93.32%|93.36%|**92.38%**|89.67%|**88.55%**|**86.58%**|**82.55%**|**75.84%**|
> | 8-Gating |92.62%|92.78%|91.25%|**89.75%**|88.53%|86.11%|81.36%|67.62%|
>
> | Lin. Mod. | s=40% | s=50% | s=60% | s=70% | s=80% | s=85% | s=92% | s=94% | s=96% |
> |-----------|-------|-------|-------|-------|-------|-------|-------|-------|-------|
> | 2-Gating |**1.184**|**1.190**|1.195|1.224|1.274|1.300|3.134|3.620|3.913|
> | 4-Gating |1.237|1.201|**1.165**|**1.129**|**1.101**|1.101|**1.816**|**2.358**|**2.938**|
> | 6-Gating |1.260|1.211|1.174|1.136|1.105|**1.082**|1.833|2.387|2.940|
> | 9-Gating |4.072|3.468|2.863|2.258|1.653|1.351|2.473|3.123|3.474|
> |10-Gating |4.209|3.791|3.494|3.237|3.036|2.945|2.842|3.235|3.600|
> |12-Gating |4.207|3.629|2.984|2.339|1.695|1.372|2.032|3.008|3.454|
>
> ------
>
> **Depth ablation for numerical instability.**
>
> Note that the plot shows the jagged “zig-zag” regularization path for large $D$ much better. The table contains RMSE values, and bold values indicate instability.
>
> | Lin. Mod. | λ=0.001 | λ=0.01 | λ=0.03 | λ=0.07 | λ=0.1 | λ=0.2 | λ=0.3 | λ=0.4 | λ=0.6 | λ=0.8 | λ=1 |
> |----------------|---------|--------|--------|--------|-------|-------|-------|-------|-------|-------|-------|
> | 2-Gating  | 2.920   | 1.845  | 1.383  | 1.206  |1.172|1.198 | 1.312 | 1.462 | 1.798 | 2.153 | 2.510 |
> | 4-Gating | 2.700   | 1.839  | 1.333  | 1.101  |1.074|1.075 | 1.105 | 1.145 | 1.249 | 1.395 | 1.922 |
> | 6-Gating | 2.652   | 1.863  | 1.381  | 1.105  | 1.073  |1.057|1.071 | 1.089 | 1.135 | 1.197 | 1.280 |
> |**9-Gating** | 8.682   | 7.704  | 6.645  | 4.759  | 4.375  |**1.048**|2.819 | 2.824 |**1.086**|**div.**| 2.859 |
> |**10-Gating**| 8.535   | 7.773  | 6.821  | 4.870  | 4.418  | 3.686 | 3.173 |**2.853**|3.499 |**2.840**| 3.510 |
> |**12-Gating**| 2.569   |**1.980**|7.091  | 4.960  | 4.484  | 3.790 |**1.050**|**1.055**|3.496 | 3.500 | 3.505 |
>
> ------
>
> # Weaknesses
>
> ### **(W1)**: restricted model/data scope
>
> While we agree that large-scale LLM or ImageNet experiments would strengthen our empirical evaluation, we argue that given the decision space {architecture types}x{data modalities}x{sparsity use cases} is quite large, we tried to best represent this in our selection of experiments, especially given our contribution has a methodological focus and the experiments serve to showcase its versatility. Our selection spans fully-connected DNNs, CNNs, neural trees, and transformers, using numerical, image, text, and speech data of various sizes, and successfully demonstrates D-Gating for input sparsity, filter sparsity, neural tree sparsity,  and sparse attention heads.
>
> ### **(W2)**: baselines are not SOTA
>
> We agree that MP and NS are solid but dated methods, particularly for transformer-based architectures where alternatives like SparseGPT exist. However, our contribution is mainly methodological and proposes a differentiable formulation of $L_{2,2/D}$ regularization, with the experimental evaluation demonstrating its effectiveness for various architectures and data modalities. SparseGPT, for example, is a method that efficiently solves a weight reconstruction problem post-training under sparsity constraints, and is thus fundamentally different from our sparse training method. Therefore, we chose to select more directly comparable baselines for these experiments, i.e., direct optimization of the non-smooth $L_{2,1}$ regularizer (besides Network Slimming and LassoNet/HSIC for the feature selection experiments), to better illustrate our method’s advantage over the naive formulation, instead of trying to establish SOTA performance for every application.
>
> ### **(W3)**: Theo. speed-up unverified in practice
>
> We again agree with you that these experiments would be interesting. But as a submission that proposes mainly a methodological contribution and empirically validates it on a variety of applications, measuring acceleration other real hardware is an interesting avenue for future research.
>
> ### **(W4)**: Missing ablation studies on runtime/memory/parameter overhead and instability)
>
> Please see our new experiments listed above on all these items.
>
> ------
>
> We hope this adequately answers your questions and that the additional results on the computational/memory/parameter overhead, instability, and performance for larger $D$.
>
> References
>
> [1] Hu, Yaohua, et al. "Group sparse optimization via $\ell_{p,q}$ regularization." Journal of Machine Learning Research 18.30 (2017): 1-52.

---

> ### Comment · Reviewer_dzXs · 2025-08-04
>
> A choice of $D$ equal to 3 or 4 appears to yield good results without adding significant computational cost, especially at high batch sizes. The author addressed my questions satisfactorily with the new, simple experiments. I understand that not all experiments can be conducted due to the exponential scaling, but hopefully we will see them in future work. I am increasing my rating to 6

---

> > ### Author Response · Authors · 2025-08-08
> >
> > Dear Reviewer dzXs,
> >
> > We sincerely thank you again for your constructive feedback and encouraging evaluation. We appreciate your acknowledgement that our additional experiments satisfactorily addressed your questions, and thank you for subsequently raising the score to 6. We believe your comments were very helpful in improving several aspects of our work.

---

### Official Review · Reviewer_YPR6 · 2025-07-04

**Clarity:** 3
**Significance:** 3
**Originality:** 3
**Rating:** 4
**Confidence:** 3

**Summary:**

The paper proposed an SGD-compatible and  framework-agnostic approach for group-lasso-like objectives without abandoning standard optimizers. The framework is based on a drop-in re-parameterization that splits every weight group into a vector  and D−1 scalar gates. The claim is training the enlarged parameter set with ordinary weight-decay (an L2 penalty) is proved to be exactly equivalent—at the level of stationary points—to training the original network with the non-smooth group-lasso-like penalty.

**Questions:**

Does the surrogate–objective equivalence still hold when switching from vanilla SGD with embedded weight-decay to common decoupled optimizers such as AdamW? A formal analysis— or at least an empirical study—of whether D-Gating remains theoretically sound or practically stable under AdamW-style decay is missing.

**Ethical Concerns:**

["NO or VERY MINOR ethics concerns only"]

**Final Justification:**

The

**Limitations:**

• Scalable experimental results are required ( ImageNet, mid-size LLMs)

•  The authors are encouraged to discuss (or provide empirical evidence) for stability and compute overhead when D>2.

• All proofs use continuous-time gradient flow. The paper will be strengthen if it extends the theory to discrete SGD (possibly under a bounded-step condition).

• Several more recent pruning baselines are missing, specially for transformers (e.g., LLM-Pruner, SparseGPT).

**Paper Formatting Concerns:**

The score was increased based on additional experimental results provided.

**Quality:**

3

**Strengths And Weaknesses:**

Strengths:
+ The authors prove that replacing each weight group w_j by a depth-D gated factorization converts plain L2 weight-decay into an exact surrogate for the non-differentiable L2,2/D  group norm.

+ Four different forms of structured sparsity (feature selection, CNN filter pruning, Transformer-head pruning, tree selection in NODEs) are studied.


Weaknesses:
-	Evaluation scope is limited to small and medium-scale benchmarks.

-	Numerical stability degrades beyond depth 4. Also, deeper gating could yield poorer optima.

---

> ### Author Rebuttal · Authors · 2025-07-29
>
> Dear Reviewer YPR6,
>
> We thank you for your time and for your encouraging comments, as well as for the interesting questions you posed. Below, respond to the raised limitations and respond to the questions point-by-point.
>
> # Questions
>
> ### **(Q1)**: Does the equivalence of objectives still hold when $L_2$ regularization is not hard-coded into the objective but a decoupled weight decay, as in AdamW, is applied?
>
> Thank you for this question. Our proposed method turns a non-smooth objective with structured sparsity regularization into a fully differentiable surrogate with $L_2$ regularization. Therefore, the explicit $L_2$ penalty is necessary for our main equivalence result in terms of minima. This equivalence is baked into the objective structure and does not rely on a specific optimizer. Consequently, we may attempt to use (S)GD, Adam, or even AdamW, as long as the objective is the gated objective with $L_2$ penalty.
> Your question, if understood correctly, however, is rather if the equivalence holds if we additionally **replace** the $L_2$ penalty in our differentiable $D$-Gating objective by decoupled weight decay (WD) in AdamW, similar to how we can exchange explicit $L_2$ regularization using WD updates in (S)GD. As updates using decoupled WD do not correspond to updates under $L_2$ regularization, the straightforward answer is (likely) no.
> However, although no exact equivalence to an $L_{2,2/D}$ regularizer holds, it is plausible that sparsity-promoting effects also emerge for AdamW, but less cleanly. Supporting this, [1] introduce decoupled WD for $\ell_q$ quasinorms and successfully train sparse DNNs with AdamW. How $L_2$ regularization and decoupled WD interact is a non-trivial question and an active area of research (see, e.g., [2]). In Appendix E, the authors of [3] discuss links between AdamW and $L_2$ regularization under certain simplifications, suggesting qualitatively similar dynamics to SGD/Adam could manifest.
>
> # Limitations:
>
> ### **(L1)/(L4)**: no large-scale experiments/SOTA baselines.
>
> While we agree that large-scale experiments would strengthen our evaluation, we argue that our contribution has a methodological focus, and the experiments serve to showcase versatility. Given the large decision space {architecture types}x{data modalities}x{sparsity use cases}, we tried to best represent this in our experiments, spanning dense DNNs, CNNs, neural trees, and transformers; using numerical, image, text, and speech data, and successfully demonstrating $D$-Gating for input/filter/tree/attention sparsity.
>
> Regarding baselines, we agree that some are solid but dated methods, particularly for transformers where alternatives like SparseGPT exist, a method that efficiently solves a weight reconstruction problem post-training under sparsity constraints, and is thus fundamentally different from our method. Hence, we chose more directly comparable baselines, e.g., direct optimization of the non-smooth $L_{2,1}$ regularizer, to better illustrate our method’s advantage over the naive formulation, instead of trying to establish SOTA performance for every application.
>
> ### **(L2)**: No overhead/stability/depth ablation experiments.
>
> We agree that those experiments would improve our submission. Thus we included several additional experiments, measuring param./memory overhead, runtime, num. stability and ablations for large $D$.
> We show that the **param./memory overhead is negligible** for all models, while **runtime increases slightly with $D$ for some models** but becomes irrelevant for larger batches. Large **$D>4$ does not improve performance but can become unstable**. We summarize the new results plots below as a table:
>
> ------
> - **Parameter count overhead for $D$-Gating (factor=new params/old params-1)**
>
> | Model | Params| Add. Params | Factor ($D=4$)| Details|
> |------------------|--------------------|---------------------------|------------------|--------|
> | Lin. Mod. | 200 |40⋅(D-1) | $6.0 \times 10^{-1}$  | 40 groups of 5 features    |
> | LeNet-300-100 | ≈266k | input dim⋅(D-1)  | $8.8 \times 10^{-3}$  | Input dim $\in [400, 784]$  |
> | VGG-16 | ≈15mio. | 4,224⋅(D-1) | $8.4 \times 10^{-4}$   | Sum of conv. filters |
> | NanoGPT | ≈10.8mio.|72⋅(D-1)| $2 \times 10^{-5}$ | 6 layers, 12 heads |
> ------
> - We also measured the **memory overhead** of $D$-Gating, finding that the utilization over all settings are almost **indiscernible from the vanilla memory usage** (see **reply to ``dzXs``** for table).
> ------
> - **Wall-clock runtime (training) for neuron, filter, and attention sparsity. Means and std. over ten runs are shown**
>
> | ms/sample| D | BS=128  | BS=256 | BS=512| BS=1024 |
> |----------------|---|----------------|----------------|----------------|----------------|
> |**LeNet-300-100**|1|0.0131 (0.0003)| 0.0072 (0.0002)| 0.0046 (0.0003)| 0.0030 (0.0003)|
> | | 2 | 0.0151 (0.0003)| 0.0082 (0.0002)| 0.0051 (0.0004)| 0.0032 (0.0000)|
> | | 3 | 0.0158 (0.0002)| 0.0093 (0.0008)| 0.0050 (0.0001)| 0.0046 (0.0011)|
> |   | 4 | 0.0170 (0.0015)| 0.0090 (0.0002)| 0.0052 (0.0001)| 0.0041 (0.0009)|
> |**VGG-16** | 1 | 0.3111 (0.0089)| 0.3054 (0.0126)| 0.1587 (0.0062)| 0.0796 (0.0038)|
> | | 2 | 0.3213 (0.0071)| 0.3066 (0.0066)| 0.1576 (0.0039)| 0.0835 (0.0027)|
> | | 3 | 0.3098 (0.0073)| 0.3019 (0.0013)| 0.1542 (0.0035)| 0.0804 (0.0016)|
> | | 4 | 0.3159 (0.0167)| 0.3057 (0.0084)| 0.1573 (0.0041)| 0.0796 (0.0021)|
>
> | ms/sample | D | BS = 16 | BS = 32 | BS = 64 |
> |--------|---|-----------|------|-------|
> |**NanoGPT**| 1 | 0.0508 (0.0012)| 0.0266 (0.0006)| 0.0174 (0.0004)|
> |  | 2 | 0.0603 (0.0011)| 0.0314 (0.0006)| 0.0185 (0.0002)|
> || 3 | 0.0606 (0.0009)| 0.0314 (0.0007)| 0.0186 (0.0002)|
> | | 4 | 0.0617 (0.0017)| 0.0317 (0.0009)| 0.0187 (0.0002)|
>
> ------
> - We further included **ablations** on how the gating depth $D$ influences performance and stability for an input-sparse LeNet-300-100, VGG-16, and a group-sparse linear model (contained in **reply to ``dzXs``** in "Lin. Mod." table). In line with the sparsity regularization literature,  $L_{2,q}$ penalties are usually only helpful up to $q=0.5$ or $q=2/3$, corresponding to $D=3,4$. In our experiments, we observe **no performance gains for large $D$**.
>
> **Ablation on depth $D$ at various sparsity levels s (test acc.)**
>
> | LeNet (ISOLET) | s=40% | s=60% | s=80% | s=90% | s=92% | s=94% | s=96% | s=98% |
> |-------|-------|-------|-------|-------|-------|-------|-------|-------|
> | 2-Gating | 95.00%| 93.66%| 68.53%| 49.81%| 46.07%| 42.32%| 38.23%| 21.04%|
> | 3-Gating | 94.89%|**94.97%**|**94.81%**|**94.81%**|93.95%|92.56%|87.28%|70.57%|
> | 4-Gating |**95.09%**|94.59%|94.27%|93.30%|**93.40%**|**92.75%**|**89.57%**|74.57%|
> | 8-Gating | 94.75%|94.09%|93.42%|92.81%|92.49%|92.32%|87.90%|**76.39%**|
> |10-Gating | 94.54%|93.76%|92.99%|92.76%|92.64%|92.24%|88.76%|73.74%|
> |20-Gating | 94.24%|93.51%|92.78%|92.53%|92.11%|92.19%|86.28%|61.31%|
>
> | VGG16 | s=40% | s=60% | s=80% | s=90% | s=92% | s=94% | s=96% | s=98% |
> |----------|-------|-------|-------|-------|-------|-------|-------|-------|
> | 2-Gating | 93.11%| 92.96%| 90.74%| 88.24%| 86.54%| 84.20%| 78.62%| 47.54%|
> | 3-Gating |**93.58%**|**93.39%**|91.55%|88.14%|87.10%|84.57%|82.04%|73.55%|
> | 4-Gating |93.32%|93.36%|**92.38%**|89.67%|**88.55%**|**86.58%**|**82.55%**|**75.84%**|
> | 8-Gating |92.62%|92.78%|91.25%|**89.75%**|88.53%|86.11%|81.36%|67.62%|
> ------
> - Due to character restrictions, further depth ablations and **instability results** for the linear model are contained in the **reply to ``dzXs``** ("Lin. Mod." tables). We find that **large $D$ exhibits increased instability but no performance benefit**.
> ------
> ### **(L3)**: Proofs use only continuous-time gradient flow. The paper will be strengthen if it extends the theory to discrete SGD.
>
> We agree that discrete-time results would strengthen our analysis beyond the existing (S)GD result in Lemma 4. We have revised the Lemma to derive the discrete-time imbalance dynamics and **prove exponential decay for sufficiently small steps**:
>
> > Consider the $D$-Gated objective Eq. (5). Then, under (S)GD with learning rate $\eta>0$, (i) the pair-wise imbalances in Eq. (13) evolve as $\mathcal{I}\_{j,d,d'}^{(t+1)} = \big(1 - \frac{4\lambda}{D} \eta\big) \mathcal{I}\_{j,d,d'}^{(t)} + \eta^2 \Delta\_{j,d,d'}^{(t)}, \\:d \neq d', d,d' \in [D]$, with different second-order terms $\Delta\_{j,d,d'}$ for $\mathcal{I}\_{j,1,d'}$ and $\mathcal{I}\_{j,d,d'}$ with $d,d'>1$. In particular, for sufficiently small $\eta$, $\mathcal{I}_{j,d,d'}^{(t+1)} \approx (1-4 \lambda \eta/D) \cdot  \mathcal{I}\_{j,d,d'}^{(t)}$ exhibits discrete exponential decay. Moreover, (ii) for balanced zero gating parameters, $(\boldsymbol{\omega}_j^{(t)},\{\gamma\_{j,d}^{(t)}\}\_{d=1}^{D-1})=\boldsymbol{0}$, the future iterates are confined to $\mathbf{w}_j^{(t')}=\boldsymbol{0} \, \forall t'>t$, and (iii), balancedness, i.e.,  $\mathcal{I}\_{j,d,d'}^{(t)}=0$, is conserved $\forall t'>t$ between any two scalar gates $\gamma\_{j,d}^{(t)},\gamma\_{j,d'}^{(t)}$.
>
> - Besides, we also include an experiment using Adam instead of SGD in Fig. 3, showing imbalance also decays to zero (even faster than SGD), although the decay is not exponential. This suggests that **Adam also leads to balancedness** and a sparsity-inducing objective. The feasibility of sparse DNN training using Adam is also demonstrated in our language experiment.
>
> ------
>
> We hope this adequately answers your questions and that the additional new results on the computational overhead, instability, and performance for larger depths, as well as our new theoretical discrete-time analysis of the imbalance evolution, further strengthen our submission in your eyes.
>
> References:
>
> [1]  Outmezguine, N.J., and Noam L.. "Decoupled Weight Decay for Any p Norm." arXiv:2404.10824 (2024).
>
> [2] Xie, S., and Zhiyuan L.. "Implicit Bias of AdamW: L∞-Norm Constrained Optimization." 41st ICML.
>
> [3] Kobayashi, S., Yassir A., and Johannes v.O.. "Weight decay induces low-rank attention layers." NeurIPS 37 (2024).

---

> > ### Comment · Reviewer_YPR6 · 2025-08-03
> >
> > Thanks for your response and additional experimental results. The score on quality was increased.

---

> > > ### Author Response · Authors · 2025-08-08
> > >
> > > Dear Reviewer YPR6,
> > >
> > > We sincerely thank you once again for your time and constructive feedback. We also appreciate your acknowledgement that our additional experiments strengthen the submission. Your comments and suggestions have been valuable in refining both the theoretical and empirical aspects of our work.

---

### Comment · Area_Chair_dx83 · 2025-08-06

Dear reviewers and authors,

The discussion period is ending soon, so please make sure to finalize the discussion (and make mandatory responses, if not completed yet). Thank you so much for your active participation!

Best,
AC.

---

### Note · Authors · 2025-08-11

Dear AC,

We sincerely thank you for handling our submission and greatly appreciate the time and effort that both you and the reviewers have invested in coordinating the process and providing constructive, insightful feedback. We are grateful that, following the rebuttal phase, all reviewers acknowledged that our rebuttals strengthened the submission and subsequently increased their ratings. In particular, we added:

- New experimental results on parameter, memory, and wall-clock runtime overheads across various settings, demonstrating our method’s negligible overhead

- New ablation studies on performance and stability for larger $D$ across different models, data modalities, and sparsity types

- An extension of our theoretical results from continuous- to discrete-time dynamics

In our rebuttals, we addressed each specific question in detail (including clarifying that our method contains explicit $L_2$ regularization, which Adam with decoupled weight decay does not replicate) and explained why domain-specific SOTA comparisons and large-scale LLM experiments are beyond the scope of our methodologically focused paper. Instead, we emphasized that the goal of our experiments is to demonstrate our method’s versatility by covering a broad range of architectures, tasks, and sparsity types.

We are genuinely grateful to the reviewers for their valuable feedback, which we believe has markedly strengthened our submission. Our rebuttal comprehensively addressed the reviewers’ questions, and we are encouraged that this was reflected in consistently increased ratings across reviews, whether in overall or quality/clarity scores. Thanks to the reviewers’ constructive suggestions, we were able to incorporate additional experiments, analyses, and clarifications that further improved the work.

We hope the AC will take into account the breadth of improvements made during our rebuttal, the incorporation of requested additional experiments and theoretical results, and the consistently positive feedback from reviewers on our rebuttals.

$~$

Best regards,

Author(s)

---

### Decision · Program_Chairs · 2025-09-17

**Decision:**

Accept (spotlight)

**Comment:**

This paper proposes D-Gating, a reparametrization for structured sparsity which keeps differentiability intact while being equivalent to group penalties. Reviewers found the proposed method to be a simple and versatile algorithm with a solid theoretical grounding. Experimental validations covers diverse scenarios, with some limitations in scale and up-to-date baselines. The authors responded effectively with additional experiments, including analysis on computational overhead and stability under deeper gating. Overall, the manuscript is technically strong, novel, and potentially impactful. Thus I recommend acceptance.